# Exploring State-Space Models for Data-Specific Neural Representations

**Jinsung Lee & Suha Kwak**
Graduate School of AI
Pohang University of Science and Technology
{jinsunglee, suha.kwak}@postech.ac.kr

## Abstract

This paper studies the problem of data-specific neural representations, aiming for compact, flexible, and modality-agnostic storage of individual visual data using neural networks. Our approach considers a visual datum as a set of discrete observations of an underlying continuous signal, thus requiring models capable of capturing the inherent structure of the signal. For this purpose, we investigate state-space models (SSMs), which are well-suited for modeling latent signal dynamics. We first explore the appealing properties of SSMs for data-specific neural representation and then present a novel framework that integrates SSMs into the representation pipeline. The proposed framework achieved compact representations and strong reconstruction performance across a range of visual data formats, suggesting the potential of SSMs for data-specific neural representations.

## 1 Introduction

Recent years have witnessed growing interest in overfitting a neural network to a single visual datum such as image (Dupont et al., 2021; Strümpler et al., 2022), video (Chen et al., 2021; Mentzer et al., 2022), or 3D instance (Martin-Brualla et al., 2021; Zhang et al., 2020). This *data-specific neural representation* paradigm, prevalent in implicit neural representations (INRs) (Sitzmann et al., 2020) and neural compression (Ballé et al., 2016; Cheng et al., 2020), aims to directly encode a datum into an embedding or the weights of a compact neural model. Such a paradigm not only serves as an effective data compression method but also offers a standardized data format that can accommodate various modalities for future neural network training (Dupont et al., 2022), with some approaches further enabling downstream applications such as spatial/temporal super-resolution (Chen et al., 2022b), denoising (Xu et al., 2022), and in/outpainting (Skorokhodov et al., 2021; Chen et al., 2023).

The central objective of the data-specific neural representations is to represent a single datum with minimal parameter complexity and maximal reconstruction quality. One of the effective strategies to achieve this comes from the recognition that visual data are essentially arrays of pixels sampled at discrete intervals from continuous signals (Sitzmann et al., 2020; Xu et al., 2022; Tancik et al., 2020; Saragadam et al., 2023). The core idea behind this approach is to project input data onto a set of established basis functions and only save their coefficients, so that the coefficients reconstruct not only the input but also the continuous signal from which the input is sampled. Although this concept has served as a fundamental principle for effective compression and reconstruction of visual data (Cooley et al., 1969; Richardson, 2011), modern approaches to data-specific neural representation do not take it into account due to the lack of well-established neural network architectures that incorporate the concept; they have instead focused merely on coordinate-to-RGB mapping (Martin-Brualla et al., 2021; Strümpler et al., 2022), bit-level quantization (Xu et al., 2018; Gordon et al., 2023) or improving the capacity of conventional neural networks to implicitly manage input redundancies (Zhou et al., 2018; Li et al., 2018).

Recently, the rise of state-space models (SSMs) has opened a new pathway to this challenge, as SSMs provide a framework for modeling continuous signals in a way that aligns with the objectives of compact neural representations. To be specific, the hidden state of SSM was initially designed to represent the coefficients that reconstruct observed data using a set of orthogonal polynomial bases (Gu et al., 2020; 2022b), which generalizes to the traditional compression algorithms. Although

the design of SSMs has become more implicit (Gu et al., 2021b; Smith et al., 2022; Gu et al., 2021a), such that their hidden states no longer explicitly represent coefficients of such continuous bases, they still retain the desirable properties necessary for effective signal modeling (Gu et al., 2021b; Guo et al., 2025; Rao, 1987; Rao & Arun, 1992), so it is worth exploring their applications.

Driven by this motivation, we explore the potential for incorporating SSMs within data-specific neural representations. We investigate the effectiveness of SSMs in compressing input data and capturing underlying signal structures, and empirically demonstrate their benefits in enhancing the reconstruction quality. However, a naïve application of SSMs presents key challenges: (1) they primarily operate on 1D sequence inputs, necessitating unnatural scanning for multi-dimensional data, and (2) they inherently preserve input sequence length, which makes them unsuitable for an effective compression method. To address these limitations, we propose *structured state-space kernel* (S3K), which distills the expressive power of SSMs into convolutional kernels. We design the kernel parameters in a way that the convolution output matches the hidden state representation of SSMs, effectively preserving their reconstruction capability. Through seamless integration with convolution, it naturally processes multi-dimensional inputs while inherently enabling expressive downsampling. In summary, our contribution is three-fold as follows:

- We for the first time investigate the integration of state-space models (SSMs) into data-specific neural representation frameworks, providing a theoretical background that explains their potential benefits in improving both expressive power and efficiency.
- We introduce S3K, an SSM-derived convolutional kernel that inherits the expressive power of SSMs while mitigating their limitations for multi-dimensional processing and downsampling.
- Our framework shows promising results across diverse visual data reconstruction tasks—images (Kodak (Kodak, 1993), CLIC2020 (Toderici et al., 2020)), videos (Bunny (Roosendaal, 2008), UVG (Mercat et al., 2020), DAVIS (Perazzi et al., 2016)), and 3D objects (Objaverse (Deitke et al., 2023))—highlighting its potential for advancing data-specific neural representations.

## 2 RELATED WORK

**State-space models.** SSMs are a family of sequence-to-sequence models that embeds historical data in state-space representation, using differential equations that involve hidden states and sequential inputs. HiPPO (Gu et al., 2020), an early state-space model, treats an input sequence as samples taken from a continuous function. This function is then approximated using a predefined set of orthogonal polynomials, with their coefficients being dynamically updated by the incoming sequential inputs. LSSL (Gu et al., 2021b) generalizes HiPPO by replacing HiPPO parameters into learnable ones, while still retaining its ability to continuously remember and store the history of observed tokens. It has developed into S4 (Gu et al., 2021a; 2022a; Nguyen et al., 2022) and S5 (Smith et al., 2022), which addresses technical inefficiencies of the previous work. Recently, Mamba (Gu & Dao, 2023) has been introduced as a state-space model that adapts its parameters based on the input sequence. Although originally designed for sequential inputs, this model has inspired various adaptations across different visual perception tasks, including images (Zhu et al., 2024a; Hu et al., 2024; Ruan & Xiang, 2024; Zhu et al., 2024b; Nguyen et al., 2022), videos (Li et al., 2025; Yang et al., 2024; Chen et al., 2024; Li et al., 2024), and 3D scenes (Liang et al., 2024; Xing et al., 2024). Although these efforts extend SSMs to multi-dimensional inputs, they primarily target sequence modeling tasks such as classification or sequence-to-sequence translation. In contrast, our work investigates SSMs as compact representations through compression of input data, a perspective that has received comparatively less attention.

**Implicit neural representations.** INRs aim at constructing a model that effectively captures continuous signals, including 3D scenes (Park et al., 2019; Mildenhall et al., 2021), images (Strümpler et al., 2022; Guo et al., 2023) and videos (Zhang et al., 2021; Chen et al., 2021). INR typically represents a continuous signal by parameterizing a field, which involves mapping between the coordinate space and the signal space. The emergence of INRs has rapidly advanced the field of data-specific neural representations, offering promising avenues for efficient compression (Strümpler et al., 2022), continuous signal modeling (Martin-Brualla et al., 2021; Chen et al., 2022a), and task-specific adaptation (Pumarola et al., 2021; Chen et al., 2021). Given this shared objective of data-specific modeling, we evaluate our method on a standard INR benchmark, highlighting its potential as a new architectural direction within the INR paradigm.

**Neural compression.** This line of work enables neural networks to learn compact representations of images and videos by incorporating advanced techniques such as entropy modeling (Ballé et al., 2018; Cheng et al., 2020) or quantization (Yang et al., 2020a;b). Early methods introduce an autoencoder-style architecture (Ballé et al., 2016; Mentzer et al., 2018), where the input is encoded into a compressed latent vector and subsequently reconstructed by the decoder. Our method adopts a similar encoder-decoder formulation while introducing state-space models (SSMs) as a new architectural component for learning compact representations, highlighting the unexplored potential of SSMs in neural compression.

## 3 PRELIMINARY: STATE-SPACE MODEL

SSM is a function that maps a 1D input signal $\phi(x)$ to a 1D output signal $y(x)$ of the same length through the latent state $h(x) \in \mathbb{C}^N$ based on the following linear differential equation:

$$
\begin{aligned}
h'(x) &= \mathbf{A}h(x) + \mathbf{B}\phi(x), \\
y(x) &= \mathbf{C}h(x),
\end{aligned}
\tag{1}
$$

where $\mathbf{A} \in \mathbb{C}^{N \times N}$ is the state transition matrix, and $\mathbf{B} \in \mathbb{C}^N$ and $\mathbf{C} \in \mathbb{C}^N$ are projection parameters. Solving the linear differential equation (1) to explicitly express $h$ yields [1]:

$$
h(x) = \int_0^x e^{(x-\tau)\mathbf{A}}\mathbf{B}\phi(\tau)d\tau \in \mathbb{C}^N,
\tag{2}
$$

where each $h_k(x) \in \mathbb{C}$ corresponds to:

$$
h_k(x) = \int_0^x e^{(x-\tau)\mathbf{A}_k}\mathbf{B}\phi(\tau)d\tau = \left\langle \phi(\tau), \overline{e^{(x-\tau)\mathbf{A}_k}\mathbf{B}} \right\rangle_{[0,x]} := \left\langle \phi(\tau), \xi_k(\tau, x) \right\rangle_{[0,x]}.
\tag{3}
$$

Here, $\langle \cdot, \cdot \rangle_{[x_1, x_2]}$ is a complex function space inner product in the given domain $[x_1, x_2]$. Intuitively, Eq. (3) tells that the $k$-th element of the hidden state $h_k(x)$ is a projection of the input $\phi(\tau)_{\tau \leq x}$ onto the function $\xi_k$. Gu et al. (2020; 2021b; 2022b;a) have established that through appropriate parameterization of the matrix $\mathbf{A}$, $\{\xi_k\}$ can serve as a set of basis functions, which enable the model to effectively capture and retain key information from the entire sequence history up to the current position $x$. This ability to project input signals onto a learned basis makes SSMs a natural fit for compression, since such a capability to model signal representation becomes beneficial. Various ways to parametrize $\mathbf{A}$ have been explored: HiPPO (Gu et al., 2020), diagonal plus low rank (Gu et al., 2021a), and diagonal (Gupta et al., 2022; Gu et al., 2022a). Among these, using a diagonal parameterization of $\mathbf{A}$ has gained popularity for its easier formulation while maintaining expressivity (Smith et al., 2022; Gu & Dao, 2023).

## 4 EXPLORING SSMS FOR DATA-SPECIFIC NEURAL REPRESENTATIONS

This section delves into advantages and proper architecture designs of data-specific neural representations using SSMs. First, we examine how SSMs encode input data and highlight their effectiveness in data-specific neural representation. Then, through an extensive experimental analysis, we identify the key characteristics of network architectures incorporating SSMs for this purpose.

### 4.1 WHAT DO SSMS ENCODE?

To understand the operational principles of SSMs and interpret their features, we bring up a classical signal processing task, *sinusoid problem*, which aims at estimating parameters of sinusoidal signals that make up the input signal. Given an input function $\phi(t)$ that takes 1D coordinate $t \in [0, L]$, we are interested in finding $\theta_n(t)$ and $c_n$ such that

$$
\phi(t) = \sum_{n=1}^N c_n e^{i\theta_n(t)},
\tag{4}
$$

---

[1]See Appendix A.1 for details.

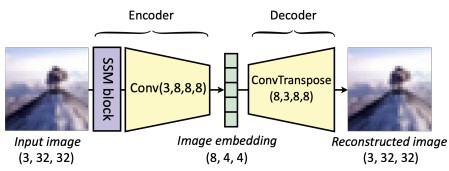

Figure 1: Baseline architecture incorporating SSMs for image reconstruction

Table 1: Image reconstruction quality in PSNR of different architectures incorporating various SSMs. (a), (b), and (c) indicate the encoder variants shown in Fig. 2.

| SSM Block | Baseline | (a) | (b) | (c) |
|---|---|---|---|---|
| Transformer (Vaswani et al., 2017) | 24.75 | 24.87 | 23.96 | 24.67 |
| S4 (Gu et al., 2021a) | 25.79 | 24.65 | 24.68 | 25.82 |
| S4D (Gu et al., 2022a) | 25.49 | 24.99 | 25.48 | 26.06 |
| S4ND (Nguyen et al., 2022) | 26.00 | 25.25 | 25.75 | 26.61 |
| S5 (Smith et al., 2022) | 25.76 | 24.84 | 24.90 | 26.44 |
| Mamba (Gu & Dao, 2023) | 24.90 | 24.82 | 24.78 | 26.58 |

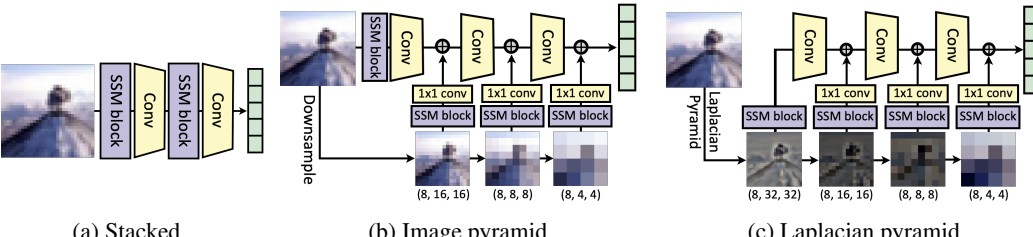

(a) Stacked      (b) Image pyramid      (c) Laplacian pyramid

Figure 2: Encoder variants incorporating SSMs for image reconstruction

where $e^{i\theta_n(t)}$ is the $n$-th sinusoidal basis and $c_n$ is its coefficient. Note that this form generalizes to various sinusoidal transformation methods, *e.g.*, setting $\theta_n(t) = -2\pi nt/L$ leads to the choice of bases used in discrete Fourier transform (Cooley et al., 1969). Under a proper choice of $\theta_n(t)$, estimating the parameter $c_n$ that approximates the input signal $\phi(t)$ offers an effective method for compression.

Interestingly, the SSM formulation allows the input signal to be decomposed into a sinusoidal form:

**Theorem 4.1.** *Let* $\mathbf{A}$ *be diagonalizable over* $\mathbb{C}$ *with non-zero distinct eigenvalues* $\{\lambda_i\}$*. Given* $\mathbf{A}$*,* $\mathbf{B}$*, and the hidden state* $h$ *computed by Eq. (2), there exists a function* $f : (\mathbf{A}, \mathbf{B}) \mapsto \mathbf{F} \in \mathbb{C}^{N \times N}$ *with which one can decompose the input function* $\phi(t)$ *as a linear combination of complex exponentials:*

$$\phi(t) = \sum_{n=1}^{N} c_n \overline{e^{\lambda_n(L-t)}}, \tag{5}$$

*where* $c_n$ *is the* $n$*-th element of* $f(\mathbf{A}, \mathbf{B})\overline{h}$*.*

The conclusion of the theorem implies that the SSM parameters $\mathbf{A}$ and $\mathbf{B}$ inherently capture signal characteristics of the input, as well as the hidden state $h$. This highlights the unique capability of SSMs: they are particularly favorable for reconstructing the input since they encode the input function $\phi(t)$ itself, unlike traditional data-specific neural representation frameworks that were originally designed to capture specific patterns or semantics.

To verify the effectiveness of SSMs in the context of compact data-specific neural representation, we conduct input reconstruction experiments where the input is compressed into an embedding and then reconstructed by a lightweight decoder, which allows us to directly assess how well the encoder captures and retains the information of the input in a compact form. To this end, we first design a simple encoder-decoder architecture that can naturally incorporate various SSMs (Fig. 1). The encoder consists of an SSM block computing the output signal by Eq. (1) and a single convolutional layer for downsampling the signal, while the decoder is composed of a deconvolutional layer (Noh et al., 2015) for upsampling. Note that the SSM block is attached directly onto the raw input and that the downsampling operation of the encoder is essential for compression due to the length-preserving nature of SSMs (Sec. 3). We consider the established SSM architectures, S4 (Gu et al., 2021a), S4D (Gu et al., 2022a), S4ND (Nguyen et al., 2022), S5 (Smith et al., 2022) and Mamba (Gu & Dao, 2023), as candidates for the SSM block.

For evaluation, we train and evaluate this baseline architecture coupled with the diverse SSMs on 1K randomly sampled images from CIFAR-100 (Krizhevsky et al., 2009). Each model is trained for 300 epochs on each image, adhering to the prevailing practice in data-specific neural representations that focuses on optimizing a lightweight model to facilitate overfitting on a single sample (Dupont et al., 2021; Chen et al., 2021). Since the SSMs are designed to operate on 1D sequences (except for

S4ND (Nguyen et al., 2022)), we preprocess the input images by flattening the $32 \times 32$ pixel grid before passing it through the SSMs. We report experimental results using the baseline architecture in the 'Baseline' column of Table 1. For comparison, we adopt a Transformer (Vaswani et al., 2017) in place of the SSM block, as it is a widely used architecture for data-specific representation (Yan et al., 2024; Mentzer et al., 2022; Liu et al., 2023; Lu et al., 2021) and has demonstrated strong performance. 'Transformer' indicates a single multi-head attention layer used in this context. The results show that, under the baseline architecture, every SSM consistently outperforms the transformer in reconstruction quality. This finding is *not* trivial, particularly considering that transformers are widely recognized for their superior performance when operating on short token sequences (Gu & Dao, 2023). This suggests that for input reconstruction, the transformer's ability to compute semantic relationships between tokens is less beneficial than in other tasks, while the input function modeling property of SSMs proves to be more advantageous, as discussed in Sec. 3.

### 4.2 Exploring architectures incorporating SSMs

To explore encoder architectures that better leverage SSMs, we experiment with several design variants. Given our focus on evaluating how well the encoder compresses the input into an embedding, we maintain a fixed decoder architecture across all configurations. We first evaluate a stacked architecture where SSM and convolutional layers alternate to form a deep network (Fig. 2(a)), and observe consistent performance drop for SSM models (Table 1(a)). We hypothesize that this decline stems from the way SSMs encode input features (Sec. 4.1): since SSMs project the input onto implicitly parameterized basis functions and stacking them results in multiple layers of such projections, repeated projection amplifies artifacts and limits the achievable reconstruction rate, analogous to generation loss from information theory (Cover, 1999).

To address this, we introduce an 'Image pyramid' variant (Fig. 2(b)), where SSMs are applied at multiple resolutions of the input. This approach improves performance (Table 1(b)), as it enables the use of representations across different scales and increases model capacity without having to stack SSM blocks. While Mamba (Gu & Dao, 2023) shows a slight drop in this setting, the overall trend confirms the benefit of incorporating SSMs across multiple resolutions.

We further explore a 'Laplacian pyramid' variant (Fig. 2(c)), a widely used decomposition method in traditional compression techniques (Burt & Adelson, 1987; Richardson, 2011). Since a Laplacian pyramid introduces less redundancy across scales than the 'Image pyramid' variant, the separate SSM blocks can be more effectively utilized. Results in Table 1(c) show consistent gains, with SSM-based models benefitting the most.

From these image reconstruction experiments, we outline key insights on incorporating SSMs: (1) Stacking SSMs in the encoding process does not yield effective results, (2) attaching SSMs to downsampled images to provide intermediate multi-scale features proves advantageous, (3) employing Laplacian pyramid decomposition further enhances performance.

## 5 Proposed Method

This section presents our method for data-specific neural representations using SSMs. We introduce our novel module, structured state-space kernel, which addresses the two major limitations of applying SSMs to neural representations of visual data: (1) their design for one-dimensional signals, which does not align directly with visual data, and (2) their inability to compress input sequences due to their length-preserving nature, which typically necessitates additional components for downsampling. Our module overcomes these challenges by leveraging structured kernels derived from SSMs, enabling efficient encoding and reconstruction of visual data.

### 5.1 Structured state-space kernel

To implement the continuous-time dynamics of SSMs on a discrete sequence, the state update of Eq. (1) is often approximated on discretized intervals using a step size parameter $\Delta$. For instance,

one can apply the hidden state update between $h(x_i)$ and $h(x_{i-1})$ using the Euler method:

$$\begin{aligned}
h(x_i) &\approx h(x_{i-1}) + \Delta h'(x_{i-1}) \\
&= h(x_{i-1}) + \Delta(\mathbf{A}h(x_{i-1}) + \mathbf{B}\phi(x_i)) \\
&:= \bar{\mathbf{A}}h(x_{i-1}) + \bar{\mathbf{B}}\phi(x_i).
\end{aligned} \tag{6}$$

Depending on the choice of the discretization method and $\Delta$, the way $\bar{\mathbf{A}}$ and $\bar{\mathbf{B}}$ are constructed may vary. Eq. (6) can be expressed as a convolution, where the hidden states evolve according to:

$$h_{-1} = 0, \quad h_0 = \bar{\mathbf{A}}h_{-1} + \bar{\mathbf{B}}\phi_0 = \bar{\mathbf{B}}\phi_0, \quad h_1 = \bar{\mathbf{A}}h_0 + \bar{\mathbf{B}}\phi_1 = \bar{\mathbf{A}}\bar{\mathbf{B}}\phi_0 + \bar{\mathbf{B}}\phi_1, \quad \cdots$$

$$h_{L-1} = \bar{\mathbf{A}}^{L-1}\bar{\mathbf{B}}\phi_0 + \bar{\mathbf{A}}^{L-2}\bar{\mathbf{B}}\phi_1 + \cdots + \bar{\mathbf{A}}\bar{\mathbf{B}}\phi_{L-2} + \bar{\mathbf{B}}\phi_{L-1}$$

$$= \left[\bar{\mathbf{A}}^{L-1}\bar{\mathbf{B}} \ \bar{\mathbf{A}}^{L-2}\bar{\mathbf{B}} \ \cdots \ \bar{\mathbf{A}}\bar{\mathbf{B}} \ \bar{\mathbf{B}}\right] \left[\phi_0 \ \phi_1 \cdots \phi_{L-1}\right]^\top, \tag{7}$$

where we denote $h(x_i)$ and $\phi(x_i)$ as $h_i$ and $\phi_i$, respectively, for brevity. Given that the last hidden state is the projection of the entire sequence onto the basis functions defined by $\mathbf{A}$ and $\mathbf{B}$ (Sec. 3), some previous work (Gu et al., 2020; 2022b) have demonstrated that the input signal can be reconstructed solely from the last hidden state. Inspired by this idea, we employ $h_{L-1}$ for downsampling, as it effectively compresses the input into a compact representation while allowing for an efficient operation by skipping the computation of intermediate hidden states, *i.e.*, $\{h_0, \ldots, h_{L-2}\}$. Once we construct the convolutional kernel $\mathbf{K}$ so that

$$\mathbf{K} = [\bar{\mathbf{A}}^{L-1}\bar{\mathbf{B}} \quad \bar{\mathbf{A}}^{L-2}\bar{\mathbf{B}} \quad \cdots \quad \bar{\mathbf{A}}\bar{\mathbf{B}} \quad \bar{\mathbf{B}}] \in \mathbb{C}^{N \times L \times C}, \tag{8}$$

we can convolve this kernel to obtain the compressed input. We refer to this kernel as *structured state-space kernel* (S3K), as it is factorized by leveraging the state transition matrix $\bar{\mathbf{A}}$ and the projection matrix $\bar{\mathbf{B}}$. In practice, we adopt the diagonal parameterization of $\mathbf{A}$ (Gu et al., 2022a; Gupta et al., 2022) to ease the computation of the power terms and adopt multi-input multi-output (MIMO) framework (Smith et al., 2022) by letting $\bar{\mathbf{B}} \in \mathbb{C}^{N \times C}$ to handle $C$ channels of input sequence simultaneously. The convolution using this structured kernel acts as a lossy compression mechanism, theoretically allowing the reconstruction of the original input signal using the learned parameters.

**Theorem 5.1.** *Let $\mathbf{A}$ be diagonalizable over $\mathbb{C}$ with non-zero distinct eigenvalues $\{\lambda_i\}$, and $\Delta$ be the step size used for the discretization of $\mathbf{A}$. Given the final hidden state $h \in \mathbb{C}^N$ after applying S3K, there exists a function $R : (\mathbf{A}, \mathbf{B}, h) \mapsto \mathbf{H} \in \mathbb{C}^{1 \times N}$ with which one can reconstruct the input sequence as:*

$$R(\mathbf{A}, \mathbf{B}, h) \left[e^{\lambda_i(L\Delta - k\Delta)}\right]_{\substack{i=1,2,\ldots,N \\ k=1,2,\ldots,L}}. \tag{9}$$

This finding further supports the interpretation of S3K as a lossy compression mechanism, where the transformed representation retains sufficient information to reconstruct the original input signal through the learned state-space parameters.

## 5.2 EXTENSION TO MULTI-DIMENSIONAL S3K

We extend S3K to multiple dimensions via outer products of multiple independent 1D S3Ks. This follows naturally from the definition of $n$D basis functions as outer products of 1D basis functions in continuous space (Cheney, 1986; Nguyen et al., 2022). The resulting $n$D kernel $\mathbf{K}$ now has dimensions $(L^{(1)}, L^{(2)}, \cdots, L^{(n)}, N, C)$, where $\{L^{(i)}\}_{i=1,\cdots,n}$ are spatial dimensions of the kernel, and $N$ and $C$ represent output and input channel dimensions, respectively. This formulation enables $n$D convolution operations on multi-dimensional inputs using structured kernels, akin to the traditional convolution.

## 5.3 ENHANCING EXPRESSIVITY

While the structured kernel theoretically finds effective basis functions and their corresponding coefficients that represent the input data, practical implementation reveals limited expressivity due to the small number of learnable parameters. To address this, we introduce several modifications to enhance the power of our model as follows.

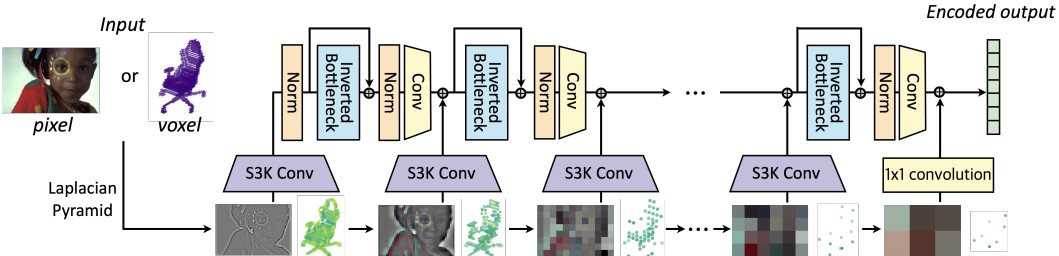

Figure 3: Structure and operation of the proposed LPNet-S3K architecture

- Input-adaptive **B**: Instead of using a fixed kernel, we adopt an adaptive mechanism (Chen et al., 2020; Gu & Dao, 2023) where the kernel parameters depend on the input, allowing dynamic adjustments to diverse signals.

- Real-valued SSM parameters: To improve numerical stability and expressivity, we follow the real parameterization of **A** and **B**, which has shown strong empirical performance in Mamba (Gu & Dao, 2023).

- Subsequent $1 \times 1$ convolution layer: We integrate a $1 \times 1$ convolution layer to further enhance the representation capacity while allowing the state size $N$ to differ from the output channel size, providing greater flexibility in model architecture design.

These modifications refine S3K into a flexible neural network module with stronger representation capacity, allowing S3K to be effectively integrated into data-specific neural representation frameworks. Additional implementation details can be found in Appendix A.8.

## 5.4 MODEL ARCHITECTURE

We illustrate the architecture of our final model in Fig. 3. We follow the design choices of the last encoder variant in Sec. 4.2 using multi-scale signals decomposed by Laplacian pyramid (Fig. 2(c)), and attach an S3K convolution layer, instead of an SSM block and the following convolutional layer, to each level of the Laplacian pyramid. We make two key modifications to complete our model: (1) replacing the intermediate MLP blocks with *inverted bottleneck* layers (Liu et al., 2022), a more advanced module that has shown superior performance across various domains (Woo et al., 2023; Chen et al., 2023; Zhao et al., 2024), and (2) using SiLU (Elfwing et al., 2018) activation and RMSNorm (Zhang & Sennrich, 2019) that have been frequently used in SSMs (Gu & Dao, 2023; Nguyen et al., 2022; Smith et al., 2022). We call this network *Laplacian Pyramid Network with S3K* (LPNet-S3K) for ease of reference.

## 6 EXPERIMENTS

To evaluate the effectiveness of the proposed LPNet-S3K architecture, we overfit the network to individual visual inputs across diverse data formats, including image, video, and 3D object. We first validate the efficacy of the LPNet and S3K architectures in the context of data-specific neural representations by evaluating their performance on images and 3D objects. Then, we evaluate our model on video INR benchmarks (NeRV), demonstrating its efficacy on the standard literature benchmarks.

**Datasets.** We evaluate our models on standard image and video reconstruction benchmarks. For images, we use Kodak (Kodak, 1993) and CLIC2020 (Toderici et al., 2020), consisting of high-resolution photographs. For videos, we follow common NeRV benchmarks, including Bunny (Roosendaal, 2008), UVG (Mercat et al., 2020), and DAVIS (Pont-Tuset et al., 2017). For 3D data, we randomly sample 1K furniture objects from Objaverse (Deitke et al., 2023). To enable compatibility with our framework, we voxelize each point cloud into a binary voxel grid, forming a cube shaped tensor that encodes the object's geometry. We describe details of each dataset in Appendix A.9.1.

Table 2: Comparison between different architectures on Kodak, CLIC2020, and Objaverse. Results are reported in PSNR/MS-SSIM.

| Method | 2D images | | 3D points |
|---|---|---|---|
| | Kodak | CLIC2020 | Objaverse |
| ConvNeXt (Liu et al., 2022) | 25.99/0.8830 | 24.39/0.8280 | 17.17/0.7536 |
| LPNet-Conv | 27.44/0.9132 | 25.41/0.8505 | 17.67/0.7815 |
| LPNet-Mamba | 27.51/0.9227 | 26.16/0.8694 | 17.74/0.7732 |
| LPNet-S3K (Ours) | 28.09/0.9331 | 26.33/0.8692 | 18.34/0.8492 |

Table 3: Comparison with the existing NeRV methods on Bunny with different model sizes. Results are reported in PSNR.

| Model size | 0.35M | 0.75M | 1.5M | 3.0M |
|---|---|---|---|---|
| HNeRV (Chen et al., 2023) | 30.15 | 32.81 | 35.57 | 37.43 |
| DNeRV (Zhao et al., 2023) | 30.15 | 33.30 | 35.22 | 38.09 |
| DS-NeRV (Yan et al., 2024) | 31.20 | 33.82 | 36.44 | 38.65 |
| SNeRV (Kim et al., 2024b) | 30.88 | 33.25 | 36.76 | 39.64 |
| Ours-S | 32.93 | 35.74 | 37.83 | 39.99 |

Table 4: Comparison with the existing NeRV methods on UVG. '*' indicates results reproduced by official codebases.

| Method | Size | Beauty | Bosp. | Honey. | Jockey | Ready. | Shake. | Yacht | Avg. |
|---|---|---|---|---|---|---|---|---|---|
| HNeRV (Chen et al., 2023) | 3.0M | 33.58 | 34.73 | 38.96 | 32.04 | 25.74 | 34.57 | 29.26 | 32.70 |
| *DNeRV (Zhao et al., 2023) | 3.4M | 34.12 | 35.65 | 39.22 | 33.72 | 28.22 | 34.80 | 29.74 | 33.64 |
| *PNeRV (Zhao et al., 2024) | 3.3M | 34.18 | 35.56 | 39.80 | 31.51 | 25.94 | 35.30 | 30.27 | 33.22 |
| DS-NeRV (Yan et al., 2024) | 3.0M | 33.97 | 35.22 | 39.56 | 32.86 | 27.10 | 35.04 | 29.40 | 33.31 |
| *SNeRV (Kim et al., 2024b) | 3.0M | 33.76 | 35.66 | 38.44 | 33.78 | 26.57 | 35.11 | 29.65 | 33.28 |
| Ours-S | 3.0M | 34.04 | 36.32 | 39.51 | 31.80 | 27.92 | 35.54 | 30.47 | 33.66 |
| Ours-P | 3.3M | 34.22 | 36.54 | 38.71 | 35.40 | 29.31 | 35.85 | 30.50 | 34.36 |

Table 5: Decoding speed comparison on UVG

| Method | PSNR | Decoding Speed | |
|---|---|---|---|
| | | sec/vid (↓) | FPS (↑) |
| HNeRV | 32.70 | 1.74 | 344.83 |
| Ours-H | 32.79 | 1.74 | 344.83 |
| PNeRV | 33.22 | 1.99 | 301.10 |
| Ours-P | 34.35 | 1.99 | 301.10 |
| SNeRV | 33.28 | 10.08 | 59.52 |
| Ours-S | 33.66 | 10.08 | 59.52 |

Table 6: Comparison with the existing NeRV methods on DAVIS. '*' indicates results reproduced by the official codebase.

| Method | Size | Bike-packing | Blackswan | BMX-trees | Breakdance | Camel | Car-rndabt | Car-shdw | Cows | Dance-twirl | Dog | Avg. |
|---|---|---|---|---|---|---|---|---|---|---|---|---|
| HNeRV (Chen et al., 2023) | 3.0M | 30.55 | 30.35 | 29.98 | 30.45 | 26.71 | 28.61 | 31.11 | 24.60 | 28.60 | 31.04 | 29.20 |
| DNeRV (Zhao et al., 2023) | 3.4M | 30.24 | 30.92 | 29.63 | 30.88 | 27.38 | 29.35 | 31.95 | 24.88 | 29.13 | 31.32 | 29.57 |
| *PNeRV (Zhao et al., 2024) | 3.3M | 28.57 | 29.17 | 28.77 | 29.67 | 27.89 | 28.76 | 31.02 | 24.39 | 28.16 | 30.95 | 28.74 |
| DS-NeRV (Yan et al., 2024) | 3.0M | - | 32.55 | 29.76 | 32.21 | 27.26 | 29.48 | 35.88 | 25.08 | 28.79 | 33.29 | - |
| SNeRV (Kim et al., 2024b) | 3.0M | 33.29 | 33.83 | 31.65 | 31.40 | 28.68 | 31.27 | 35.79 | 25.14 | 30.41 | 34.11 | 31.56 |
| Ours-S | 3.0M | 34.33 | 34.58 | 31.72 | 33.86 | 30.04 | 32.35 | 36.69 | 26.47 | 30.25 | 33.21 | 33.86 |
| Ours-P | 3.3M | 32.15 | 34.17 | 32.48 | 33.15 | 29.04 | 32.38 | 32.35 | 25.94 | 30.94 | 33.97 | 32.25 |

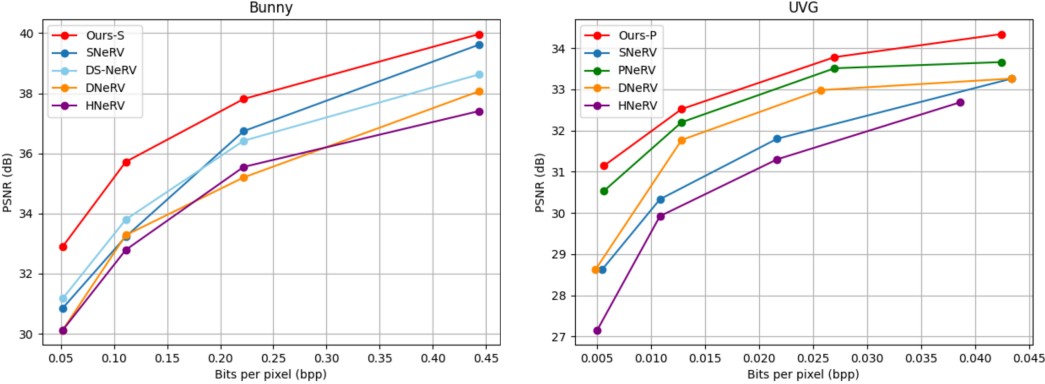

Figure 4: Rate-distortion plot on Bunny (left) and UVG (right).

**Evaluation metrics.** We adhere to the standard evaluation protocols, reporting Peak Signal-to-Noise Ratio (PSNR, in dB) and/or Multi-Scale Structural Similarity Index Measure (MS-SSIM) (Wang et al., 2003) as fidelity metrics across all reconstruction tasks.

**Implementation details.** For images and 3D objects, we adopt a simple setup by attaching multiple 2D or 3D deconvolutional layers (Noh et al., 2015) on top of LPNet-S3K to reconstruct the input. For videos, we replace the convolutional encoders of HNeRV (Chen et al., 2023), SNeRV (Kim et al., 2024b), and PNeRV-L (Zhao et al., 2024) with LPNet-S3K, denoted as 'Ours-H', 'Ours-S' and 'Ours-P', respectively. Additional implementation details are provided in Appendix A.9.2.

## 6.1 QUANTITATIVE COMPARISONS

**Images and 3D objects.** Quantitative results are reported in Table 2. To validate the effectiveness of LPNet (Sec. 5.4), the proposed baseline architecture, we construct a ConvNeXt (Liu et al., 2022) variant that follows the same configuration of LPNet, as ConvNeXt is a widely adopted encoder in

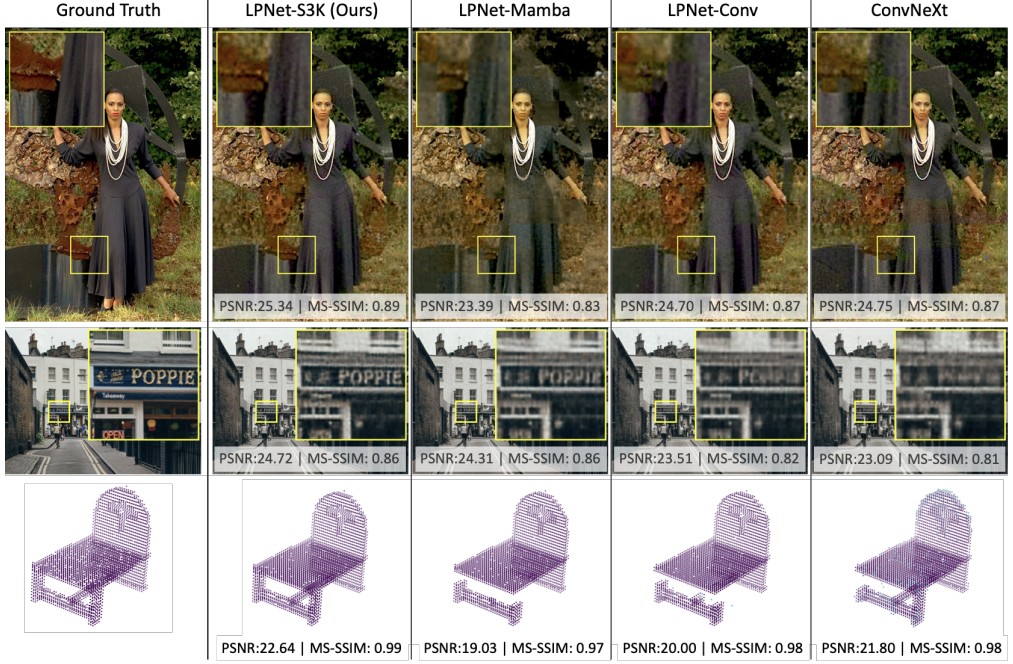

Figure 5: Reconstruction results on images (Kodak, CLIC2020) and voxelized points (Objaverse)

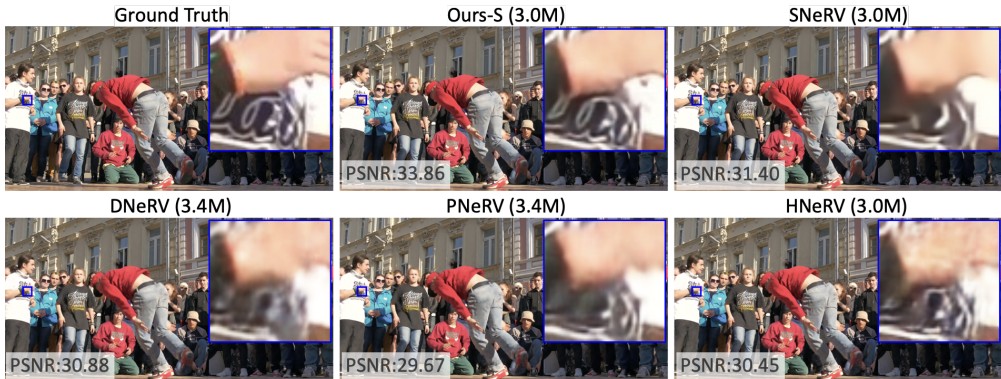

Figure 6: Reconstruction results on a DAVIS video

the literature on data-specific neural representations (Chen et al., 2023; Zhao et al., 2023; 2024; Kim et al., 2024b). For a fair comparison with ConvNeXt, we integrate standard convolutional layers into LPNet, denoted as LPNet-Conv. The performance gap between ConvNeXt and LPNet-Conv in Table 2 highlights the superiority of the LPNet architecture. We then compare LPNet-Conv and LPNet-S3K to assess the contribution of S3K convolution; LPNet-S3K outperformed LPNet-Conv across all benchmarks as shown in the table, underscoring the benefits of SSMs for data-specific neural representations. To assess the distinct advantage of S3K, we also experiment with Mamba (Gu & Dao, 2023) as an alternative SSM (LPNet-Mamba). Specifically, we switch the S3K convolutions with Mamba followed by a standard convolutional layer. LPNet-S3K outperforms LPNet-Mamba, showing its effectiveness as a compression-specialized SSM. Refer to Appendix A.9.3 for further ablation studies. It is worth noting that the approach introduced by LPNet and S3K is orthogonal to prevalent techniques in data-specific neural representations, such as bit quantization (Kim et al., 2024a; Ladune et al., 2023; Damodaran et al., 2023) or learning image priors from large-scale datasets (Ballé et al., 2016; 2018; Cheng et al., 2020; Strümpler et al., 2022; Catania & Allegra, 2023). This distinction highlights the potential complementarity of our framework: we believe that integrating LPNet-S3K with these existing techniques could further enhance performance and offer new insights into the design of compact and effective data-specific neural representations.

**NeRV benchmarks.** We evaluate our method on standard NeRV benchmarks, with results shown in Table 3 (Bunny), Table 4 and 5 (UVG), and Table 6 (DAVIS). We include convolution-based NeRV models as our baselines, following Yan et al. (2024) and Kim et al. (2024b). Additional comparisons to MLP-based methods are provided in Appendix A.9.4. On Bunny, our method ranks the best across all model sizes. On UVG, our method outperforms previous arts and even surpasses models of larger sizes: Ours-S achieves 33.66 PSNR with only 3.0M parameters, exceeding the performance of bigger models like DNeRV and PNeRV. We provide an additional rate–distortion plot in Fig. 4, where we vary the bits-per-pixel (bpp) and compare the corresponding model performance. Our model consistently outperforms the baselines across different bpp levels, demonstrating its effective compression capability across a wide range of compression ratios. It is noteworthy that our method enhances the performance *while leaving the decoder part unchanged*, ensuring the inference cost remains the same (Table 5). This aspect offers a meaningful advantage in NeRV, since video decoding speed is critical for its real-time streaming applications. On DAVIS, our model ranks either first or second across various videos, surpassing most prior methods. All results are obtained by employing existing decoders, indicating that the performance gains are entirely from our SSM-based encoder; this also implies even greater potential of our method with a dedicated decoder design.

## 6.2 QUALITATIVE ANALYSIS

We present qualitative comparisons on images and 3D objects in Fig. 5, showing reconstruction results on Kodak (Kodak, 1993), CLIC2020 (Toderici et al., 2020), and Objaverse (Deitke et al., 2023). Across all datasets, our model consistently preserves finer details, such as high-frequency textures in the background (first row) or legible text on signage (second row), and underlying structure of 3D geometry (third row). We provide video reconstruction results in Fig. 6. Our model shows superior performance despite its smaller model size compared to PNeRV and DNeRV, and effectively preserves fine details, such as a person's hand or typographies on a t-shirt. Additional qualitative results are provided in Appendix A.9.5.

## 7 CONCLUSION AND FUTURE DIRECTIONS

In this paper, we present the first attempt to link SSMs to data-specific neural representations. To this end, we explore network architectures for effective SSM integration and analyze how different architectures suit the characteristics of SSMs. As a result, we propose S3K, which harnesses the expressiveness of SSMs while enabling natural multi-dimensional data processing and downsampling. These results are theoretically supported and together lead to a novel and powerful data-specific neural representation framework. Our framework achieves strong performance across diverse visual data formats, including images, videos and 3D objects, and remains superior on challenging NeRV benchmarks despite not being designed for NeRV.

In the following, we outline potential directions to improve and extend our framework.

1. *Designing a dedicated decoder*: This work focuses primarily on designing the SSM-based encoder, while employing a simple upsampling decoder or decoders from other methods. While our results demonstrate that the encoder alone significantly contributes to the performance improvements, a decoder tailored to the characteristics of the SSM encoded features may further improve the performance of our model.

2. *Reducing encoding complexity*: Constructing an input-sized kernel using state-space model parameters leads to substantial computational overhead: about $20\times$ more memory and $4\times$ more FLOPs than a plain convolution. While effective, this approach may limit scalability. More efficient alternatives—such as avoiding explicit kernel construction through mathematically equivalent formulations (Nguyen et al., 2022; Gu et al., 2021a), or employing hardware-optimized implementations (Gu & Dao, 2023)—could alleviate this burden.

3. *Application to autoencoders*: Although our method is proposed for data-specific neural networks, its compressive property can be exploited to produce compressive representation beyond individual inputs. Note that modern autoencoders used for generative modeling often rely on convolution-based architectures (Black Forest Labs, 2023; Rombach et al., 2022) or signal processing methods (Agarwal et al., 2025). S3K, which aligns closely with both convolution and signal processing principles, could enable compressive autoencoders that encode inputs using fewer tokens.

## 8 ACKNOWLEDGEMENTS

This research was supported by the High-Performance Computing Support Project (RQT-25-070271) and the IITP grants (RS-2024-00509258, RS-2024-00469482, RS-2024-00457882, RS-2019-II191906), funded by the Government of the Republic of Korea (Ministry of Science and ICT).

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

## A APPENDIX

This material provides proofs for the theorems in the main paper (*i.e.*, Theorem 4.1 and Theorem 5.1), and additional details omitted in the manuscript due to the space constraint.

### A.1 SOLVING THE LINEAR DIFFERENTIAL EQUATION

We solve the linear differential equation of the state-space model (Eq. (1)), and derive its solution in the form of Eq. (2). As the derivation in this section depends on the original state-space formulation, we first restate Eq. (1) from the main paper for reference:

$$h'(x) = \mathbf{A}h(x) + \mathbf{B}\phi(x). \tag{10}$$

We start by solving a homogeneous first-order matrix ordinary differential equation (ODE), $h'_h(x) = \mathbf{A}h_h(x)$, which is a standard matrix ODE. Its solution is $h_h(x) = e^{\mathbf{A}x}C$, where $C \in \mathbb{C}^N$ is a constant vector determined by initial conditions. Allowing the constant $C$ to vary with $x$, *i.e.*, $C := u(x)$, derives the particular solution of the form $h(x) = e^{\mathbf{A}x}u(x)$. Plugging this to Eq. (10) yields:

$$\frac{d}{dx}e^{\mathbf{A}x}u(x) = \mathbf{A}h(x) + \mathbf{B}\phi(x) \tag{11}$$

$$\Rightarrow \mathbf{A}e^{\mathbf{A}x}u(x) + e^{\mathbf{A}x}u'(x) = \mathbf{A}e^{\mathbf{A}x}u(x) + \mathbf{B}\phi(x), \tag{12}$$

and canceling the terms gives:

$$e^{\mathbf{A}x}u'(x) = \mathbf{B}\phi(x) \tag{13}$$

$$\Rightarrow u'(x) = e^{-\mathbf{A}x}\mathbf{B}\phi(x) \tag{14}$$

$$\Rightarrow u(x) = \int_0^x e^{-\mathbf{A}\tau}\mathbf{B}\phi(\tau)d\tau + C'. \tag{15}$$

For practical implementation, since we set $h(0) = u(0) = 0$, we can set $C' = 0$. Thus, $h(x) = e^{\mathbf{A}x}u(x)$ becomes:

$$h(x) = e^{\mathbf{A}x}u(x) \tag{16}$$

$$= e^{\mathbf{A}x}\int_0^x e^{-\mathbf{A}\tau}\mathbf{B}\phi(\tau)d\tau \tag{17}$$

$$= \int_0^x e^{(x-\tau)\mathbf{A}}\mathbf{B}\phi(\tau)d\tau, \tag{18}$$

which matches the target expression (Eq. (2)).

### A.2 PROOF FOR THEOREM 3.1

**Theorem 3.1.** *Let $\mathbf{A}$ be diagonalizable over $\mathbb{C}$ with non-zero distinct eigenvalues $\{\lambda_i\}$. Given $\mathbf{A}$, $\mathbf{B}$, and the hidden state $h$ computed by Eq. (2), there exists a function $f : (\mathbf{A}, \mathbf{B}) \mapsto \mathbf{F} \in \mathbb{C}^{N \times N}$ with which one can decompose the input function $\phi(t)$ as a linear combination of complex exponentials:*

$$\phi(t) = \sum_{n=1}^N c_n \overline{e^{\lambda_n(L-t)}}, \tag{19}$$

*where $c_n$ is the $n$-th element of $f(\mathbf{A}, \mathbf{B})\overline{h}$.*

*Proof.* Since $\mathbf{A}$ is diagonalizable, we can write

$$\xi(\tau, L) = \overline{e^{(L-\tau)\mathbf{A}}\mathbf{B}} \tag{20}$$

$$= \overline{\mathbf{V}e^{(L-\tau)\mathbf{\Lambda}}\mathbf{V}^{-1}\mathbf{B}}, \tag{21}$$

for some matrix $\mathbf{V} \in \mathbb{C}^{N \times N}$ and a diagonal matrix $\mathbf{\Lambda} \in \mathbb{C}^{N \times N}$. Let $\mathbf{b} := \mathbf{V}^{-1}\mathbf{B}$, so that

$$\mathbf{V}^{-1}\mathbf{B} = \begin{bmatrix} b_1 \\ b_2 \\ \vdots \\ b_N \end{bmatrix}. \tag{22}$$

Since $e^{(L-\tau)\mathbf{\Lambda}}$ is diagonal,

$$e^{(L-\tau)\mathbf{\Lambda}}\mathbf{b} = \begin{bmatrix} e^{(L-\tau)\lambda_1}b_1 \\ e^{(L-\tau)\lambda_2}b_2 \\ \vdots \\ e^{(L-\tau)\lambda_N}b_N \end{bmatrix}, \tag{23}$$

and thus, multiplying $\mathbf{V}$ yields:

$$\mathbf{V}e^{(L-\tau)\mathbf{\Lambda}}\mathbf{b} = \mathbf{V} \begin{bmatrix} e^{(L-\tau)\lambda_1}b_1 \\ e^{(L-\tau)\lambda_2}b_2 \\ \vdots \\ e^{(L-\tau)\lambda_N}b_N \end{bmatrix} \tag{24}$$

$$= \begin{bmatrix} \sum_{n=1}^{N} v_{1n}e^{(L-\tau)\lambda_n}b_n \\ \sum_{n=1}^{N} v_{2n}e^{(L-\tau)\lambda_n}b_n \\ \vdots \\ \sum_{n=1}^{N} v_{Nn}e^{(L-\tau)\lambda_n}b_n \end{bmatrix}, \tag{25}$$

where $v_{ij}$ is the $(i, j)$-th element of the matrix $\mathbf{V}$. Hence, the $\xi_k(\tau, x)$ from Eq. (20) can be expressed as:

$$\xi_k(\tau, L) = \sum_{n=1}^{N} \overline{v_{kn}e^{(L-\tau)\lambda_n}b_n}, \tag{26}$$

$$:= \sum_{n=1}^{N} d_{kn}\overline{e^{\lambda_n(L-\tau)}} \tag{27}$$

for some constant $d_{kn}$. Hence, it becomes natural to choose sinusoidal bases $e^{i\theta_n(t)} = \overline{e^{\lambda_n(L-t)}}$ and express the input function $\phi(t)$ as:

$$\sum_{n=1}^{N} c_n \overline{e^{\lambda_n(L-t)}}. \tag{28}$$

Note that $\{\overline{e^{\lambda_n(L-t)}}\}$ consists of complex exponentials with $N$ distinct frequencies, which ensures their linear independence (Lang, 2012). Since this property allows them to serve as valid basis functions, we can obtain orthonormal basis functions $\{\psi_n(t)\}$ that span the same functional space as $\{\overline{e^{\lambda_n(L-t)}}\}$. Let the change-of-basis matrix from $\{\psi_n(t)\}$ to $\{\overline{e^{\lambda_n(L-t)}}\}$ defined as $\mathbf{P}$:

$$\mathbf{P} \begin{bmatrix} \psi_1(t) \\ \psi_2(t) \\ \vdots \\ \psi_N(t) \end{bmatrix} = \begin{bmatrix} \overline{e^{\lambda_1(L-t)}} \\ \overline{e^{\lambda_2(L-t)}} \\ \vdots \\ \overline{e^{\lambda_N(L-t)}} \end{bmatrix}. \tag{29}$$

Then, $\xi_k$ from Eq. (27) can be rephrased to:

$$\xi_k(\tau, L) = \begin{bmatrix} d_{k1} & d_{k2} & \cdots & d_{kN} \end{bmatrix} \begin{bmatrix} \overline{e^{\lambda_1(L-\tau)}} \\ \overline{e^{\lambda_2(L-\tau)}} \\ \vdots \\ \overline{e^{\lambda_N(L-\tau)}} \end{bmatrix}. \tag{30}$$

$$= \begin{bmatrix} d_{k1} & d_{k2} & \cdots & d_{kN} \end{bmatrix} \mathbf{P} \begin{bmatrix} \psi_1(\tau) \\ \psi_2(\tau) \\ \vdots \\ \psi_N(\tau) \end{bmatrix} \tag{31}$$

$$:= \sum_{n=1}^{N} g_{kn}\psi_n(\tau), \tag{32}$$

where $g_{kn}$ is the inner product between $\begin{bmatrix} d_{k1} & d_{k2} & \cdots & d_{kN} \end{bmatrix}$ and $n$-th column of $\mathbf{P}$. Similarly, we can express $\phi(t)$ from Eq. (28) with different bases, *i.e.*, $\sum_{n=1}^{N} m_n \psi_n(t)$, where $\begin{bmatrix} m_1 & m_2 & \cdots & m_N \end{bmatrix} = \begin{bmatrix} c_1 & c_2 & \cdots & c_N \end{bmatrix} \mathbf{P}$. Then, plugging $\phi(t) = \sum_{n=1}^{N} m_n \psi_n(t)$ into Eq. (3) gives:

$$h_k = \langle \sum_{n=1}^{N} m_n \psi_n(\tau), \xi_k(\tau, L) \rangle_{[0,L]} \tag{33}$$

$$= \sum_{n=1}^{N} \overline{m_n} \langle \psi_n(\tau), \sum_{j=1}^{N} g_{kj} \psi_n(\tau) \rangle_{[0,L]} \tag{34}$$

$$= \sum_{n=1}^{N} \overline{m_n g_{kn}} = \begin{bmatrix} \overline{g_{k1}} & \overline{g_{k2}} & \cdots & \overline{g_{kN}} \end{bmatrix} \begin{bmatrix} \overline{m_1} \\ \overline{m_2} \\ \vdots \\ \overline{m_N} \end{bmatrix} \tag{35}$$

$$= \begin{bmatrix} \overline{g_{k1}} & \overline{g_{k2}} & \cdots & \overline{g_{kN}} \end{bmatrix} \overline{\mathbf{P}}^T \overline{\mathbf{c}}, \tag{36}$$

where $\mathbf{c} = \begin{bmatrix} c_1 & c_2 & \cdots & c_N \end{bmatrix}^T$.

Let $\mathbf{h} := \begin{bmatrix} h_1 & h_2 & \cdots & h_N \end{bmatrix}^T$ and $\overline{\mathbf{G}} := [\overline{g_{ij}}]$. Stacking Eq. (36) yields:

$$\mathbf{h} = \overline{\mathbf{G}} \overline{\mathbf{P}}^T \overline{\mathbf{c}} \quad \Leftrightarrow \quad \overline{\mathbf{c}} = (\overline{\mathbf{G}} \overline{\mathbf{P}}^T)^{-1} \mathbf{h} \quad \Leftrightarrow \quad \mathbf{c} = (\mathbf{G} \mathbf{P}^T)^{-1} \overline{\mathbf{h}}, \tag{37}$$

which enables us to rewrite $\phi(t)$ by plugging $\mathbf{c}$ to Eq. (28). $\qquad \square$

### A.3 Proof for Theorem 4.1

**Theorem 4.1.** *Let $\mathbf{A}$ be diagonalizable over $\mathbb{C}$ with non-zero distinct eigenvalues $\{\lambda_i\}$, and $\Delta$ be the step size used for the discretization of $\mathbf{A}$. Given the final hidden state $h \in \mathbb{C}^N$ after applying S3K, there exists a function $R : (\mathbf{A}, \mathbf{B}, h) \mapsto \mathbf{H} \in \mathbb{C}^{1 \times N}$ with which one can reconstruct the input sequence as:*

$$R(\mathbf{A}, \mathbf{B}, h) \left[ e^{\lambda_i (L\Delta - k\Delta)} \right]_{\substack{i=1,2,\ldots,N \\ k=1,2,\ldots,L}} . \tag{38}$$

*Proof.* Let $T = L\Delta = x_{L-1}$. Note that

$$h = \int_0^T e^{(T-\tau)\mathbf{A}} \mathbf{B} \phi(\tau) d\tau. \tag{39}$$

The diagonalizability of $A$ gives:

$$e^{(x-\tau)\mathbf{A}} = \mathbf{V} e^{\mathbf{\Lambda}(x-\tau)} \mathbf{V}^{-1}. \tag{40}$$

Let $\mathbf{V}^{-1} \mathbf{B} := \tilde{\mathbf{B}}$ have no zero elements, then

$$\tilde{\mathbf{h}} := \mathbf{V}^{-1} h \tag{41}$$

$$= \int_0^T e^{\mathbf{\Lambda}(T-\tau)} \tilde{\mathbf{B}} \phi(\tau) d\tau \tag{42}$$

$$\Rightarrow c_k := \frac{\tilde{\mathbf{h}}_k}{\tilde{\mathbf{B}}_k} = \int_0^T e^{\lambda_k (T-\tau)} \phi(\tau) d\tau \tag{43}$$

Note that our goal is to recover $\phi(\tau)$ from $h$. This can be accomplished by finding the *dual basis function* $\{f_k(\tau)\}$ of the basis function $\{e^{\lambda_k (T-\tau)}\}$ for $k \in \{1, 2, \cdots, N\}$. The dual basis function $f_k(\tau)$ satisfies

$$\int_0^T e^{\lambda_i (T-\tau)} f_j(\tau) d\tau = \delta_{ij}, \tag{44}$$

where $\delta_{ij}$ is the Kronecker delta. It is worth to note that having $f_k(\tau)$ leads to the expression of $\phi(\tau)$ as:

$$\phi(\tau) = \sum_{k=1}^{N} c_k f_k(\tau), \tag{45}$$

which can be easily shown when plugging Eq. (45) to Eq. (43). Thus, our problem is now converted to finding the dual basis $\{f_k\}$ that corresponds to $\{e^{\lambda_k(T-\tau)}\}$. We start from expressing $f_j(\tau)$ as a linear combination of $e^{\lambda_k(T-\tau)}$:

$$f_j(\tau) = \sum_{k=1}^{N} z_{jk} e^{\lambda_k(T-\tau)} \tag{46}$$

for some $z_{jk}$. Plugging Eq. (46) to Eq. (44) gives:

$$\int_0^T e^{\lambda_i(T-\tau)} \sum_{k=1}^{N} z_{jk} e^{\lambda_k(T-\tau)} d\tau \tag{47}$$

$$= \sum_{k=1}^{N} z_{jk} \int_0^T e^{(\lambda_i+\lambda_k)(T-\tau)} d\tau = \delta_{ij} \tag{48}$$

If we let the matrix $\mathbf{Z} := \{z_{ik}\}$ and $\mathbf{G} = \{g_{ik}\} = \{\frac{e^{(\lambda_i+\lambda_k)T}-1}{\lambda_i+\lambda_k}\}$, directly solving the integral yields:

$$\sum_{k=1}^{N} z_{jk} \frac{e^{(\lambda_i+\lambda_k)T}-1}{\lambda_i+\lambda_k} = \delta_{ij} \iff \mathbf{Z}\mathbf{G} = \mathbf{I}. \tag{49}$$

Hence, we obtain $\mathbf{Z} = \mathbf{G}^{-1}$. According to Eq. (46),

$$\mathbf{f}(\tau) = \mathbf{G}^{-1} \begin{bmatrix} e^{\lambda_1(T-\tau)} \\ e^{\lambda_2(T-\tau)} \\ \vdots \\ e^{\lambda_N(T-\tau)}, \end{bmatrix} \tag{50}$$

and plugging this to Eq. (45) gives:

$$\phi(\tau) = \mathbf{c}^T \mathbf{f}(\tau) = \mathbf{c}^T \mathbf{G}^{-1} \begin{bmatrix} e^{\lambda_1(T-\tau)} \\ e^{\lambda_2(T-\tau)} \\ \vdots \\ e^{\lambda_N(T-\tau)} \end{bmatrix} \tag{51}$$

where $\mathbf{c} = \begin{bmatrix} c_1 \\ \vdots \\ c_N \end{bmatrix}$. If we put everything together,

$$\phi(\tau) = \mathbf{c}^T \mathbf{G}^{-1} \begin{bmatrix} e^{\lambda_1(T-\tau)} \\ e^{\lambda_2(T-\tau)} \\ \vdots \\ e^{\lambda_N(T-\tau)} \end{bmatrix} \tag{52}$$

$$= \left(\frac{\mathbf{V}^{-1}\mathbf{h}}{\mathbf{V}^{-1}\mathbf{B}}\right)^T \mathbf{G}^{-1} \begin{bmatrix} e^{\lambda_1(T-\tau)} \\ e^{\lambda_2(T-\tau)} \\ \vdots \\ e^{\lambda_N(T-\tau)} \end{bmatrix}. \tag{53}$$

We abused element-wise division operation here for simplicity. Evaluating $\phi(\tau)$ at $\tau \in \{0, \Delta, 2\Delta, \cdots, (L-1)\Delta\}$ completes the proof. $\qquad \square$

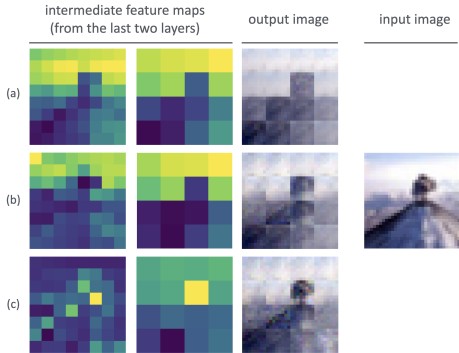

Figure A1: Feature map visualizations of three encoder variants explored in Sec 4.2. (a), (b) and (c) correspond to each variant in Fig. 2.

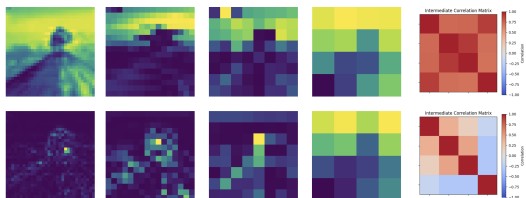

Figure A2: SSM block output of Image pyramid (top) and Laplacian pyramid (bottom) variants. The rightmost column shows correlation matrix between these maps.

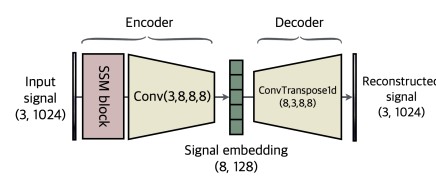

Figure A3: Baseline architecture for 1D signal reconstruction

Table A1: Performance (PSNR) of different architectures incorporating various SSMs for the 1D signal task. (a), (b), and (c) indicate the encoder variants illustrated in Fig. A4.

| SSM Block | PSNR | | | |
|---|---|---|---|---|
| | Baseline | (a) | (b) | (c) |
| Transformer (Vaswani et al., 2017) | 28.09 | 27.98 | 27.27 | 27.47 |
| S4 (Gu et al., 2021a) | 29.81 | 28.43 | 28.66 | 31.20 |
| S4D (Gu et al., 2022a) | 28.99 | 26.24 | 27.73 | 29.94 |
| S4ND (Nguyen et al., 2022) | 29.98 | 28.17 | 28.52 | 31.05 |
| S5 (Smith et al., 2022) | 28.93 | 26.39 | 28.81 | 28.98 |
| Mamba (Gu & Dao, 2023) | 30.62 | 26.78 | 27.30 | 30.73 |

## A.4 VISUAL MATERIALS TO SUPPORT EMPIRICAL FINDINGS IN SEC. 4

Figure A1 supports our hypothesis on why stacking SSM layers deteriorates the reconstruction: the feature maps from stacked SSM layers show clear block-like artifacts, and these distortions persist into the reconstructed output. The consistency between feature-level and output-level artifacts aligns with our hypothesis that stacking SSM projections accumulates errors, ultimately limiting the achievable reconstruction quality.

Fig. A2 illustrates that residual maps provide meaningfully distinct inputs to the model: in the 'Laplacian pyramid' variant, the outputs of the SSM blocks vary substantially across scales, whereas in the 'Image pyramid' variant, the outputs exhibit noticeable redundancy.

## A.5 SSM FOR 1D SIGNAL RECONSTRUCTION

As a natural extension of the reconstruction experiment in Sec. 4.2, we conduct additional studies on 1D signal reconstruction to further validate the effectiveness of SSM-based architectures in capturing signals. We follow the similar experimental setup as described in Sec. 4.2, with the primary modifications being the use of 1D convolutional layers in place of 2D ones and adjustments to the input dimensionality. Specifically, we convert input images into 1D signals by flattening them in a zig-zag manner, transforming the input shape from $(3, 32, 32)$ to $(3, 1024)$. The kernel size of the convolution is maintained to $8$, resulting in an encoded embedding that is one-eighth the length of the input. For clarity, we illustrate the modified baseline architecture and its variants in Fig. A3 and Fig. A4, respectively.

The results of the experiment are demonstrated in Table A1. We find the similar tendency we have observed in Sec. 4.2, which suggests that SSMs, when placed in an appropriate reconstruction setting, hold strong potential in signal reconstruction as implied in Sec. 4.1.

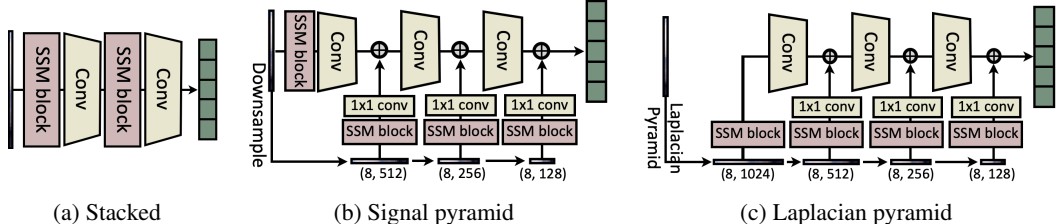

| (a) Stacked | (b) Signal pyramid | (c) Laplacian pyramid |

Figure A4: Encoder variants incorporating SSMs for 1D signal reconstruction

Table A2: Parameter count comparison on different variants of S3K 2D convolution.

| Method | subsequent $1 \times 1$ conv. | Adaptive $\mathbf{B}$ | Adaptive $\Delta$ | Real $\mathbf{A}, \mathbf{B}$ | # param | $C = 3, D = 16,$ $K = 5, N = 16$ |
|---|---|---|---|---|---|---|
| Conv2d($C$,$D$,$K$) | - | - | - | - | $CDK^2 + D$ | 3088 |
| | ✗ | ✗ | ✗ | ✗ | $6N + 4NC$ | 288 |
| | ✓ | ✗ | ✗ | ✗ | $6N + 4NC + (N+1)D$ | 560 |
| S3KConv2d | ✓ | ✓ | ✗ | ✗ | $8N + 2NK + (N+1)D$ | 656 |
| ($C$,$N$,$D$,$K$) | ✓ | ✓ | ✓ | ✗ | $8N + 2NK + K + (N+1)D$ | 664 |
| | ✓ | ✓ | ✗ | ✓ | $8N + 2NK + (N+1)D$ | 624 |
| | ✓ | ✓ | ✓ | ✓ | $6N + 2NK + K + (N+1)D$ | 632 |

Table A3: Computation cost comparison between SSM-convolution variants, measured on Kodak.

| Method | FLOPs (G) ↓ | Memory (MB) ↓ | Time (ms) ↓ | PSNR ↑ |
|---|---|---|---|---|
| S4D + Conv | **2.00** | **453.3** | 100.4 | 26.73 |
| Mamba + Conv | 2.36 | 667.3 | 94.4 | 27.51 |
| S3KConv | 3.34 | 3445.8 | **62.4** | **28.09** |

## A.6 PARAMETER COUNT ANALYSIS

As discussed in Sec. 5.3, the direct application of S3K convolution suffers from limited expressivity, primarily due to an insufficient parameter budget that constrains the network's representational capacity. Table A2 presents how each modification contributes to the parameter count of the 2D convolution using S3K. Following the standard convolutional network notation, Conv2d($C, D, K$) denotes a 2D convolutional layer that transforms an input with $C$ channels into $D$ output channels using a kernel of size $K$. Similarly, S3KConv2d($C, N, D, K$) performs the same transformation but introduces an intermediate state of size $N$ to model the structured state-space dynamics. The primary contributor to the high parameter count in standard convolutional networks is the $CDK^2$ term, which involves the multiplication of four factors and grows rapidly with channel and kernel size. In contrast, S3K layers are designed with more compact parameterization, where the largest terms involve only two multiplicative factors, resulting in significantly fewer parameters. Among all the architectural modifications, the most significant increase in parameter count arises from the subsequent $1 \times 1$ convolution, which projects the $N$-dimensional latent state into the desired size of output feature. On the other hand, the adaptivity of state-space parameters also introduces a relatively modest parameter increase, though the exact impact depends on the choice of state size $N$.

## A.7 COMPUTATIONAL COST ANALYSIS

As discussed in Sec. 7, our explicit-kernel formulation leads to additional computational burden compared to other SSM-based convolutions that avoid kernel materialization. For explicit comparison, we benchmark these approaches under the same setting by replacing the convolution in our architecture with an S4D or Mamba block followed by a standard convolution layer in Table A3. The results highlight the trade-offs: SSM-convolutions that do not materialize the kernel (*i.e.*, S4D, Mamba) consume less memory relative to our explicit kernel, but require additional runtime due to the sequential structure of their implicit state updates. Conversely, S3KConv incurs higher memory due to explicit kernel construction, but benefits from more parallel, convolution-like execution and achieves better reconstruction quality.

## A.8 MODEL IMPLEMENTATION DETAILS

This section provides additional implementation details that were omitted from the main paper due to space constraints.

### A.8.1 INITIALIZATION OF $\mathbf{A}$ AND $\mathbf{B}$

We mainly follow the initialization scheme introduced in S5 (Smith et al., 2022) and Mamba (Gu & Dao, 2023). For complex initialization of $\mathbf{A}$ and $\mathbf{B}$ for ablation, we follow HiPPO initialization of $\mathbf{A}$ and use eigenvectors $\mathbf{V}$ from diagonalization of $\mathbf{A}$ for initialization of $\mathbf{B}$, as done in S5 (Smith et al., 2022). For real implementation, we set $\mathbf{\Lambda}$, the $N$ diagonal elements of $\mathbf{A}$, be $\mathbf{\Lambda}_n = -(n+1)$, and employ normal initialization for $\mathbf{B}$.

### A.8.2 OPERATIONAL DETAILS OF S3KCONV2D LAYER

We elaborate on operational details of S3KConv2d layer, which naturally extends to S3K convolutions for $N$-dimensional inputs.

S3KConv2d layer takes the same arguments of the ordinary convolutional layer: input channel dimension $C_{\text{in}}$, output channel dimension $C_{\text{out}}$, and kernel size $(K_1, K_2)$. For simplicity, we assume trivial settings for stride, padding, and dilation. Let the input $X_0 \in \mathbb{R}^{B \times C_0 \times H_0 \times W_0}$, then we first project input to have $C$ channel dimension by $1 \times 1$ convolution: $X \in \mathbb{R}^{B \times C \times H_0 \times W_0}$. Now we construct a kernel for each spatial dimension. For clearer explanation, we focus on the input $X_w \in \mathbb{R}^{B \times C \times K_1 \times K_2}$, which represents the local window extracted during convolution. The following operations are applied in parallel across all such windows as the kernel slides over the input. Note that in Mamba (Gu & Dao, 2023), a linear layer is applied to the length-$L$ input sequence to obtain the input-adaptive $\mathbf{B}$ of length $L$. Since the kernel of $i$-th dimension needs to be a length-$K_i$ 1D kernel, we apply linear layer to the other spatial dimension. Specifically, let $\mathbf{B}_{\text{proj}}^{(i)}(d_{\text{in}}, d_{\text{out}})$ be the linear projection layer that transforms the channel dimension from $d_{\text{in}}$ to $d_{\text{out}}$ to produce $\mathbf{B}^{(i)}$, $\mathbf{B}$ used to construct the $i$-th dimension S3K 1D kernel. Then, we set $(d_{\text{in}}, d_{\text{out}}) = (K_2, N)$ for $i = 1$ and $(K_1, N)$ for $i = 2$, so that we obtain $\mathbf{B}^{(i)} \in \mathbb{R}^{B \times C \times K_i \times N}$. With $\mathbf{\Lambda}^{(i)} \in \mathbb{R}^N$ and step size $\Delta^{(i)} \in \mathbb{R}^N$, we discretize $\mathbf{\Lambda}^{(i)}$ and $\mathbf{B}^{(i)}$ using zero-order hold (ZOH) method:

$$\overline{\mathbf{\Lambda}}^{(i)} = e^{\mathbf{\Lambda}^{(i)}\Delta^{(i)}}, \qquad \overline{\mathbf{B}}^{(i)} = (\mathbf{\Lambda}^{(i)})^{-1}(\Delta^{(i)}\overline{\mathbf{\Lambda}}^{(i)} - \mathbf{I})\mathbf{B}^{(i)}, \tag{54}$$

and compute the kernel as in Eq. (7). Since we have kernel $\mathbf{K}^{(i)} \in \mathbb{R}^{B \times C \times K_i}$, we can take outer product of these kernels to construct 2D kernel $\mathbf{K} \in \mathbb{R}^{B \times C \times N \times K_1 \times K_2}$, and apply this kernel to the input $X_w$:

$$X_w^{(\text{out})} \in \mathbb{R}^{B \times N \times 1 \times 1}, \qquad \text{where } (X_w^{(\text{out})})_{bn} = \sum_c \sum_{k_2} \sum_{k_1} ((X_w)_{[b,c,k_1,k_2]})\mathbf{K}_{[b,c,n,k_1,k_2]}. \tag{55}$$

We also incorporate gating mechanisms (Gu et al., 2022a; Nguyen et al., 2022; Gu & Dao, 2023) and residual connections for complete implementation, as they have proven effective and are widely adopted as standard components in SSM block designs. Specifically, we project the initial input $X_0$ to have $N$ channels: $X_{\text{res}} \in \mathbb{R}^{B \times N \times H \times W}$, and 2D average pool with the same kernel size $(K_1, K_2)$ and the stride to produce the tensor $X_{\text{res}}^{(\text{out})}$, matching the output size of the convolution. A SiLU (Elfwing et al., 2018) activation is applied to this residual tensor, which is then used to gate the convolution output via element-wise multiplication. Finally, we add $X_{\text{res}}^{(\text{out})}$ back to the gated output to complete the residual connection.

## A.9 EXPERIMENT DETAILS

This section provides additional experiment details that were omitted from the main paper due to space constraints.

### A.9.1 DATASETS

**Kodak, CLIC2020.** The Kodak dataset (Kodak, 1993) is a set of 24 natural photographs of resolution $512 \times 768$. The CLIC2020 (Toderici et al., 2020) dataset includes 41 images of varying

Table A4: Implementation details of the experiments from Sec. 6

| Dataset | Encoder | Enc. strides | Decoder | Dec. strides | Feature dims. | Learning rate |
|---------|---------|--------------|---------|--------------|---------------|---------------|
| Kodak | ConvNeXt | [16, 4, 2, 2] | ConvTranspose2D | [4, 4, 4, 2, 2] | [64, 64, 64, 16] | 1e-2 |
| | LPNet-Conv | [16, 4, 2, 2] | ConvTranspose2D | [4, 4, 4, 2, 2] | [64, 64, 64, 16] | 3e-2 |
| | LPNet-Mamba | [32, 2, 2, 2] | ConvTranspose2D | [4, 4, 4, 2, 2] | [64, 64, 64, 16] | 1e-2 |
| | LPNet-S3K | [32, 2, 2, 2] | ConvTranspose2D | [4, 4, 4, 2, 2] | [64, 64, 64, 16] | 2e-2 |
| CLIC2020 | ConvNeXt | [4, 4, 2] | ConvTranspose2D | [4, 2, 2, 2] | [64, 64, 64, 16] | 1e-2 |
| | LPNet-Conv | [4, 4, 2] | ConvTranspose2D | [4, 2, 2, 2] | [64, 64, 64, 16] | 5e-3 |
| | LPNet-Mamba | [8, 2, 2] | ConvTranspose2D | [4, 2, 2, 2] | [64, 64, 64, 16] | 1e-2 |
| | LPNet-S3K | [8, 2, 2] | ConvTranspose2D | [4, 2, 2, 2] | [64, 64, 64, 16] | 1.6e-2 |
| Objaverse | ConvNeXt | [2, 2, 2, 2] | ConvTranspose3D | [2, 2, 2, 2] | [64, 64, 64, 16] | 3e-3 |
| | LPNet-Conv | [2, 2, 2, 2] | ConvTranspose3D | [2, 2, 2, 2] | [64, 64, 64, 16] | 3e-3 |
| | LPNet-Mamba | [4, 2, 2] | ConvTranspose3D | [2, 2, 2, 2] | [64, 64, 64, 16] | 4e-3 |
| | LPNet-S3K | [4, 2, 2] | ConvTranspose3D | [2, 2, 2, 2] | [64, 64, 64, 16] | 3e-3 |
| Bunny | LPNet-S3K | [5, 4, 4, 2, 2] | SNeRV (Kim et al., 2024b) | [5, 4, 2, 2] | [64, 64, 64, 16] | 3e-4 |
| UVG | LPNet-S3K | [10, 8, 3, 2] | HNeRV (Chen et al., 2023) | [5, 4, 4, 3, 2] | [64, 64, 64, 16] | 2e-4 |
| | LPNet-S3K | [10, 8, 3, 2] | SNeRV (Kim et al., 2024b) | [5, 4, 4, 3, 2] | [64, 64, 64, 16] | 2e-4 |
| | LPNet-S3K | [10, 8, 3, 2] | PNeRV-L (Zhao et al., 2024) | - | - | 2e-4 |
| DAVIS | LPNet-S3K | [10, 8, 3, 2] | SNeRV Kim et al. (2024b) | [5, 4, 4, 3, 2] | [64, 64, 64, 16] | 2e-4 |
| | LPNet-S3K | [10, 8, 3, 2] | PNeRV-L Zhao et al. (2024) | - | - | 2e-4 |

resolutions, allowing side lengths up to 2048 pixels. Both datasets are commonly used for image compression tasks, as they contain rich high-frequency details and complex scenes.

**Bunny, UVG, DAVIS.** Bunny (Roosendaal, 2008) is a 132-frame, animated short film, while UVG (Mercat et al., 2020) is a long 1080p video dataset comprising sequences of 300 or 600 frames. Both are widely used benchmarks for video compression. DAVIS (Huang et al., 2017) is a densely annotated featuring short 1080p video clips, commonly used for video segmentation. Since NeRV benchmarks often include a subset of DAVIS, we follow this convention and select the following video clips; 'bike-packing', 'blackswan', 'bmx-trees', 'breakdance', 'camel', 'car-roundabout', 'car-shadow', 'cows', 'dance-twirl', and 'dog'.

**Objaverse.** Objaverse is a large-scale dataset containing over 800K web-crawled 3D objects, which covers a wide range of functional categories and geometric variations. As one of the largest publicly available collections of 3D assets, it is often used for 3D understanding, neural rendering, shape reconstruction, and vision-language grounding. In our work, we take the 1K 3D objects from '*Furnitures*' subset, which consists of everyday household items such as sofas, chairs, and tables. We voxelize each furniture object to have a binary grid of size $32^3$ (approximately 33K voxels), where the voxel occupancy encodes the object's structure.

### A.9.2 IMPLEMENTATION DETAILS

We provide detailed implementation configurations in Table A4. Since the decoder architecture remains consistent across all experiments, we list the different experimental variants under the 'Encoder' column. The 'Enc. strides' column specifies the strides applied at each encoder stage, indicating the spatial downsampling factor between successive layers. Analogously, 'Dec. strides' indicates the upsampling factor at each decoder stage. PNeRV (Zhao et al., 2024) does not employ conventional upsampling methods such as deconvolution (Noh et al., 2015) or pixelshuffle (Shi et al., 2016), thus we do not specify its decoder strides. We observe that SSM-based encoders perform better when the first encoder layer uses a larger kernel size, while standard convolution-based encoders tend to perform well with comparably more uniform stride settings across layers. Accordingly, we use a larger stride in the first encoder layer for SSM-based models to match this behavior. The 'Feature dims.' column specifies the channel width at each encoder stage, which is kept consistent across all experiments. Since each model has different characteristics, we observe that they require distinct learning rates to achieve optimal performance. Therefore, we carefully search across a range of learning rates for each dataset and report the best results.

Table A5: Ablation study of LPNet-S3K components on the Bunny dataset. We employ HNeRV (Chen et al., 2023) decoder for the experiment.

| subsequent $1 \times 1$ conv. | Adaptive $\mathbf{B}$ | Adaptive $\Delta$ | Inverted Bottleneck | Real $\mathbf{A}, \mathbf{B}$ | SiLU act. | RMS norm | PSNR |
|---|---|---|---|---|---|---|---|
| ✗ | ✗ | ✗ | ✗ | ✗ | ✗ | ✗ | 36.37 |
| ✓ | ✗ | ✗ | ✗ | ✗ | ✗ | ✗ | 36.67 |
| ✓ | ✓ | ✗ | ✗ | ✗ | ✗ | ✗ | 36.92 |
| ✓ | ✓ | ✓ | ✗ | ✗ | ✗ | ✗ | 36.26 |
| ✓ | ✓ | ✗ | ✓ | ✗ | ✗ | ✗ | 36.94 |
| ✓ | ✓ | ✗ | ✓ | ✗ | ✓ | ✓ | 37.03 |
| ✓ | ✓ | ✗ | ✓ | ✓ | ✗ | ✗ | 36.92 |
| ✓ | ✓ | ✗ | ✓ | ✓ | ✓ | ✗ | 36.99 |
| ✓ | ✓ | ✗ | ✓ | ✓ | ✓ | ✓ | 37.04 |

Table A6: Ablation of state size on the Bunny dataset.

| state size | PSNR |
|---|---|
| 16 | 36.74 |
| 32 | 36.88 |
| 64 | 37.03 |
| 128 | 36.92 |

Table A7: Ablation of state size on the Kodak dataset.

| state size | PSNR |
|---|---|
| 4 | 27.65 |
| 8 | 27.98 |
| 16 | 28.09 |
| 32 | 28.04 |

Table A8: Ablation of state size on the Objaverse dataset.

| state size | PSNR |
|---|---|
| 8 | 17.56 |
| 16 | 17.98 |
| 32 | 18.34 |
| 64 | 18.29 |

### A.9.3 ABLATION STUDIES

We present an ablation study in Table A5 to evaluate the impact of the architectural modifications introduced in Sec. 5.3 and Sec. 5.4. In addition, we evaluate an alternative design inspired by Mamba (Gu & Dao, 2023), where the step size $\Delta$ is made input-adaptive. The results show that each proposed component contributes positively to performance, except for the adaptive $\Delta$: performance drops significantly from 36.92 to 36.26 when $\Delta$ is made input-dependent. We hypothesize that this is due to a mismatch between the objective of input-selectivity, a primary reason for adopting adaptive $\Delta$, and input reconstruction. Since input reconstruction demands uniform attention across all regions of the input for accurate reconstruction, the input selectivity introduced by adaptive $\Delta$ may be less effective than in other tasks. We also find that the complex initialization of $\mathbf{A}$ and $\mathbf{B}$ (Smith et al., 2022) is equally helpful, aligning with observations in Mamba (Gu & Dao, 2023) that such initialization of $\mathbf{A}$ is aids in processing continuous inputs. However, for performance optimization in terms of speed, we stick to the real parametrization.

We also provide thorough ablations on state size of S3K layers in Table A6-A8. We use a state size of 8 (images), 32 (3D data), 64 (videos), and apply the same state size across all S3K layers for each setting. Empirically, increasing the state size in the first S3K layer meaningfully enhances the performance, likely because this layer is the primary opportunity to capture high-frequency spatial details and a larger state size helps preserve this. In contrast, increasing the state size in later layers yields only marginal gains. For simplicity and consistency, we adopt the first layer's optimal state size for all layers. In our experiments, performance improves as the state size increases but begins to saturate once the state size is sufficiently large relative to the signal complexity. In practice, this occurs around 8 for images, around 32 for 3D data, and around 64 for videos–values we therefore use throughout the model.

### A.9.4 COMPARISON TO MLP-BASED METHODS

LPNet-S3K is yet inapplicable to coordinate-to-rgb mapping methods (*i.e.*, MLP-based methods) such as Liu et al. (2024); Sitzmann et al. (2020) (images) or Kwan et al. (2024); Kim et al. (2022) (videos), categorizing itself as a *convolution-based* neural representation, in contrast to other mainstream neural representation *MLP-based* neural representation (coordinate-based, INR). Hence, we have compared our methods only with convolution-based methods in Sec. 6.1, as it has been a standard practice to compare models within the same family (Kim et al., 2024b; Yan et al., 2024; Kim et al., 2022; Shin et al., 2024), due to their fundamentally different characteristics which are illustrated in Table A9.

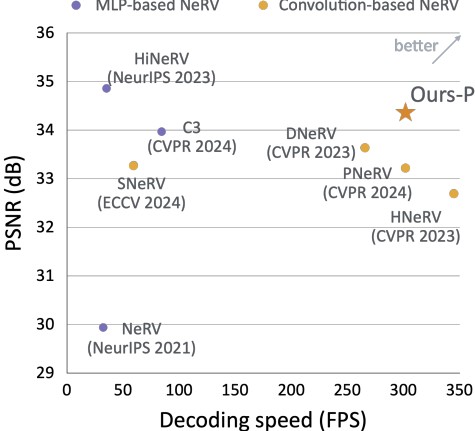

Figure A5: Reconstruction quality and decoding speed trade-off in modern NeRV models. We collect models of 0.02-0.03bpp on a 600-frame UVG video, which occupy 300 epochs to train. FPS is measured with NVIDIA A6000ada GPU.

Table A9: Comparison between MLP-based methods and convolution-based methods

| Type | recon. quality | decoding speed | Examples |
|---|---|---|---|
| MLP-based (INR) | high | low | SIREN (Sitzmann et al., 2020) FINER (Liu et al., 2024) HiNeRV (Kwan et al., 2024) NeRV (Chen et al., 2021) NVP (Kim et al., 2022) C3 (Kim et al., 2024a) |
| Convolution -based | low | high | HNeRV (Chen et al., 2023) DNeRV (Zhao et al., 2023) PNeRV (Zhao et al., 2024) DS-NeRV (Yan et al., 2024) SNeRV (Kim et al., 2024b) Ours |

Table A10: Comparison between LPNet-S3K to FINER.

| Method | Epochs | Num params | Training time (s/epoch) | Memory (MB) | Kodak | CLIC |
|---|---|---|---|---|---|---|
| LPNet-S3K | 300 | 150K | 0.05 | 35210 | 28.09 | 26.33 |
| FINER (Liu et al., 2024) | 300 | 199K | 0.4 | 4106 | 28.68 | 27.12 |

Fig. A5 also exhibits their differences clearly: convolution-based models offer very fast decoding speed while sacrificing their reconstruction quality. On the other hand, MLP-based methods in general show higher fidelity, but have slower decoding speed, which limits their real-time application. Unlike the prevailing NeRV trend of trading decoding speed for higher reconstruction quality, LPNet-S3K serves as a fidelity enhancer for convolution-based models without adding inference cost.

We also provide a comparison with MLP-based image neural representation, FINER (Liu et al., 2024) in Table A10. FINER shows better fidelity compared to LPNet-S3K, but requires much more time and memory for training. On the other hand, LPNet-S3K shows greater efficiency, while underperforming in terms of reconstruction quality.

### A.9.5 ADDITIONAL QUALITATIVE RESULTS

We provide additional qualitative results in Fig. A6 (Kodak), Fig. A7 (CLIC2020), Fig. A8 (Objaverse), and Fig. A9-A14 (DAVIS).

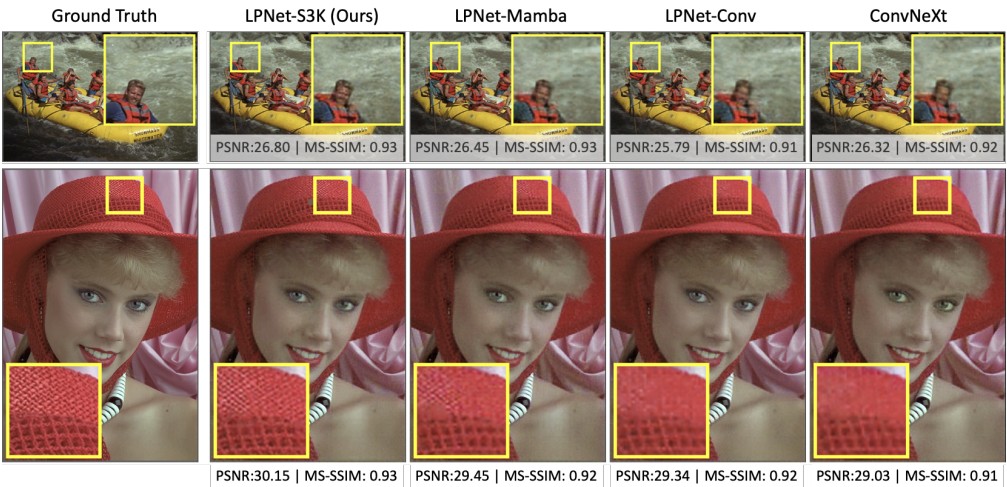

Figure A6: Reconstruction results on the Kodak dataset

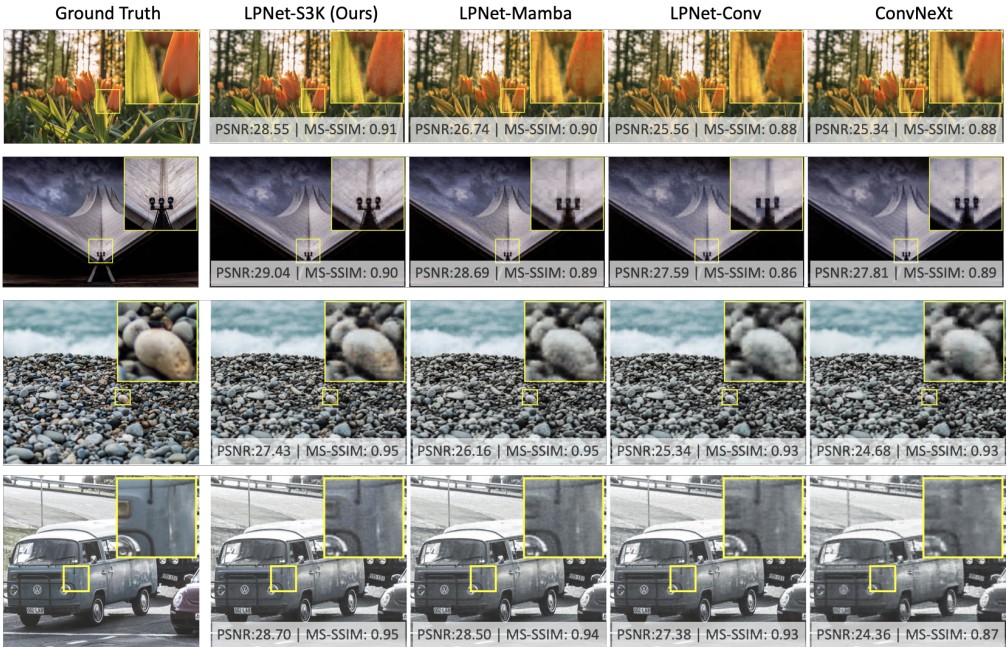

Figure A7: Reconstruction results on the CLIC2020 dataset

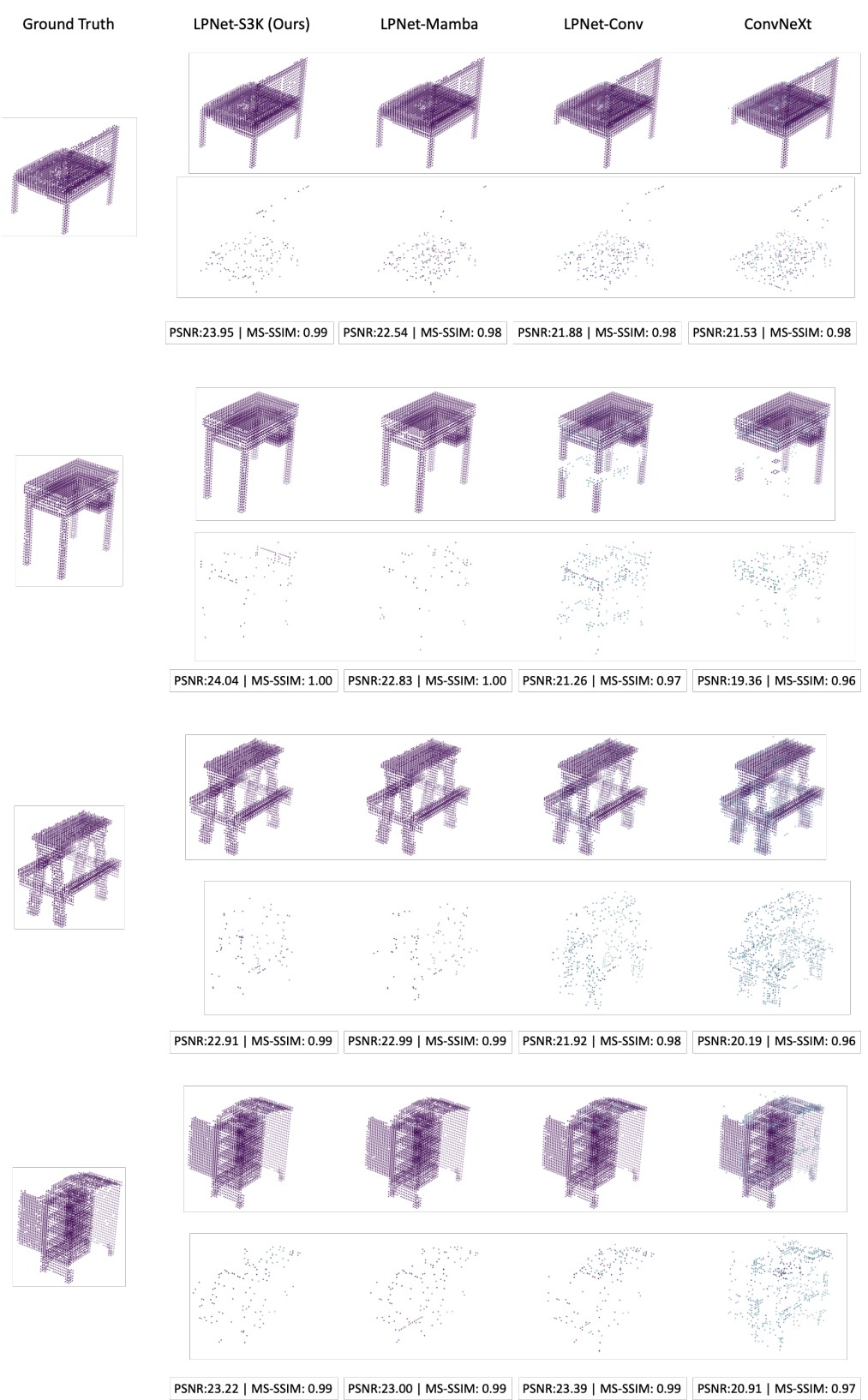

Figure A8: Reconstruction results on the Objaverse dataset. For clearer comparison, we present the difference visualization under each reconstruction result.

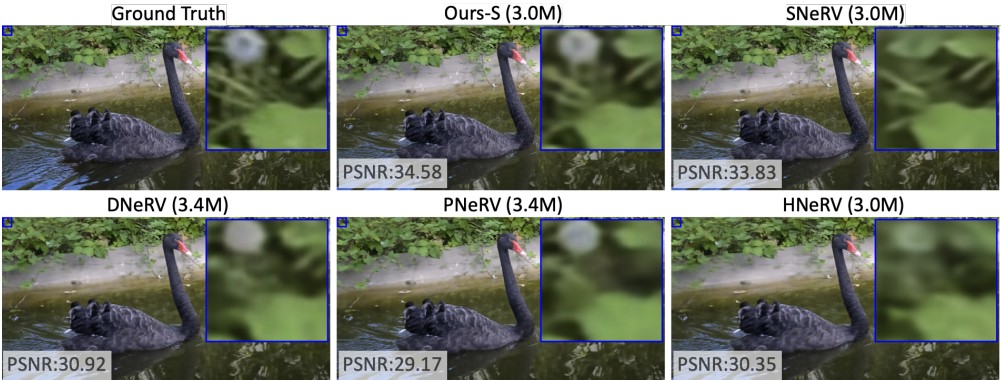

Figure A9: Reconstruction results on 'blackswan' from the DAVIS dataset

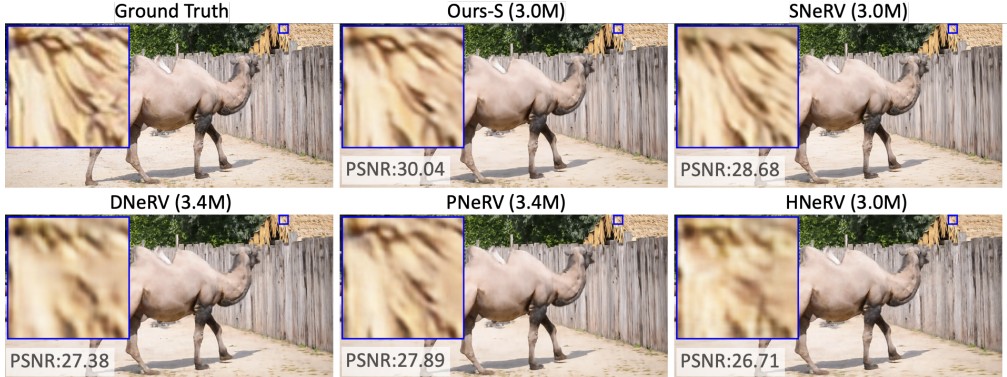

Figure A10: Reconstruction results on 'camel' from the DAVIS dataset

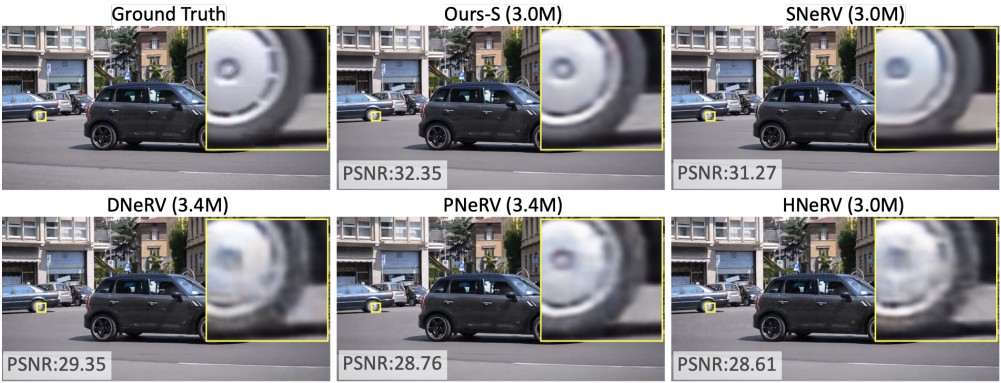

Figure A11: Reconstruction results on 'car-roundabout' from the DAVIS dataset

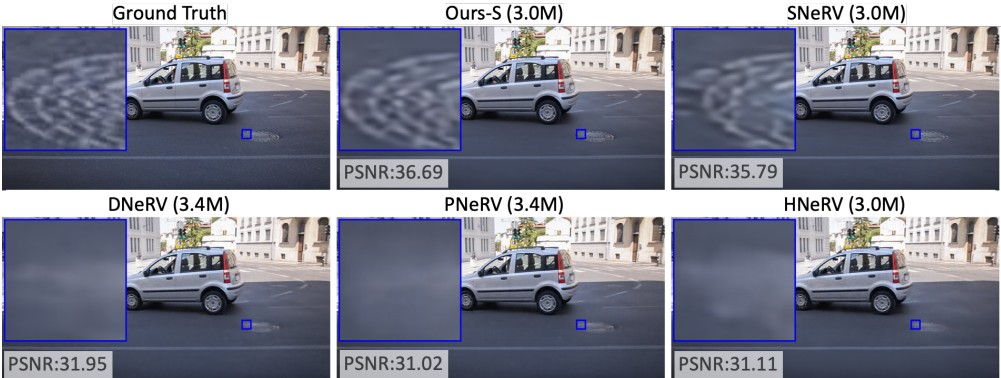

Figure A12: Reconstruction results on 'car-shadow' from the DAVIS dataset

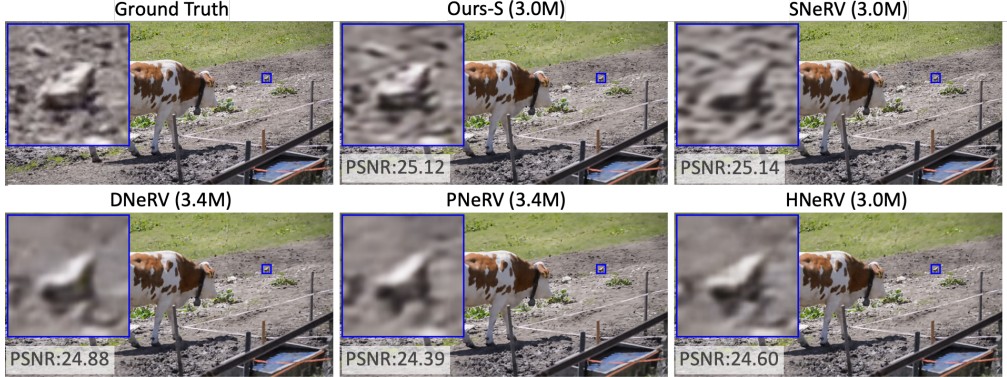

Figure A13: Reconstruction results on 'cow' from the DAVIS dataset

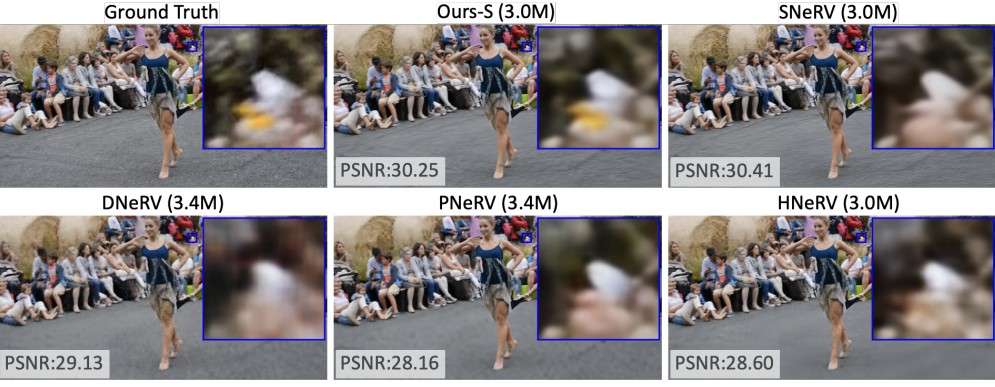

Figure A14: Reconstruction results on 'dance-twirl' from the DAVIS dataset

