# OpenReview forum: "Exploring State-Space Models for Data-Specific Neural Representations"
_ICLR.cc/2026/Conference — ICLR 2026 Poster_

### Official Review · Reviewer_My7V · 2025-10-21

**Soundness:** 2
**Presentation:** 4
**Contribution:** 2
**Rating:** 4
**Confidence:** 3

**Summary:**

This paper proposes to use state-space models (SSMs) to build neural representations. The main motivation for the use of SSMs is that many signals are discrete observations of an underlying continuous signal, permitting the use of classical signal processing techniques. In this paper, the authors extend an older approach of compressing signals by storing them as their coefficients in a fixed basis, which can be done using SSMs. This paper proposes a new SSM-based architecture which they call S3K. Based on their analysis of different architectures, this paper combines S3K with multi-scale features and a Laplacian pyramid to create a new SSM-based architecture for neural representations, which they call LPNet-S3K. The proposed method is evaluated on 2D image reconstruction, video reconstruction, and 3D shape reconstruction.

**Strengths:**

**(S1) Novelty**: this paper introduces the new idea of using SSMs to create neural representations and proposes a novel SSM-based architecture.

**(S2) Architectural analysis**: This paper analyzes different architectures to create the best architecture for SSM-based neural representations.

**(S3) Theoretical justification**: This paper also provides some theory to back up their claims.

**Weaknesses:**

**(W1) Motivation/Limited applicability**: the S3K/LPNet-S3K method is mainly evaluated on toy tasks of the following form: given a signal, reconstruct that signal. The proposed method is not evaluated for more complicated tasks involving neural representations. Potential more complex tasks that are signal-to-signal include video interpolation, video superresolution, and 3D reconstruction from 2D partial views. Additionally, it is not possible to build S3K-based neural representations that map coordinates to function values, which represent a significant class of neural representations used today.

**(W2) Computational trade-off of convolutional methods vs MLP methods**: The authors mention that, compared to convolutional methods like the one proposed in the paper, MLP methods offer higher fidelity but slower decoding speed. It seems to me that this may not matter in practice when there exist fast and efficient MLP neural representations such as InstantNGP [3].

**(W3) Experimental setup**: For the 2D image and 3D object reconstruction experiments, the only baselines are ablations of the proposed model and ConvNeXt, which was not designed as neural representation. I think the performance would be better contextualized by adding previous neural representation methods, such as FINER, which was mentioned in the Appendix A.8.4. Additionally, recent baselines for video reconstruction are either not mentioned in the main paper or not included in the paper. These include HiNeRV (only mentioned in the Appendix), *-Boost [1], NVRC [2].

**(W4) Empirical results**: The empirical justification of the method is mixed. For 2D and 3D reconstruction, the proposed method is worse than MLP methods but better than the use of a vanilla convolutional network (ConvNeXt). For video reconstruction, the method is generally better than all but HiNeRV, with some recent methods possibly missing. The increased performance comes at the cost of much more expensive training due to the increased expense of the LPNet-S3K encoder (20x more memory, 4x more FLOPS).

[1] Zhang, Xinjie, et al. "Boosting neural representations for videos with a conditional decoder." Proceedings of the IEEE/CVF Conference on Computer Vision and Pattern Recognition. 2024.

[2] Kwan, Ho Man, et al. "NVRC: Neural video representation compression." Advances in Neural Information Processing Systems 37 (2024): 132440-132462.

[3] Müller, Thomas, et al. "Instant neural graphics primitives with a multiresolution hash encoding." ACM transactions on graphics (TOG) 41.4 (2022): 1-15.

**Questions:**

**(Q1)**: How does the proposed method compare to recent methods such as Boost [1] and NVRC [2]?

**(Q2)**: How does the decoding speed of convolutional methods compare with that of efficient MLP methods such as InstantNGP [3]?

**(Q3)**: Can SSM-based methods be extended to create coordinate neural networks?

[1] Zhang, Xinjie, et al. "Boosting neural representations for videos with a conditional decoder." Proceedings of the IEEE/CVF Conference on Computer Vision and Pattern Recognition. 2024.

[2] Kwan, Ho Man, et al. "NVRC: Neural video representation compression." Advances in Neural Information Processing Systems 37 (2024): 132440-132462.

[3] Müller, Thomas, et al. "Instant neural graphics primitives with a multiresolution hash encoding." ACM transactions on graphics (TOG) 41.4 (2022): 1-15.

---

> ### Author Response · Authors · 2025-11-21
>
> We sincerely appreciate the reviewer’s effort in providing detailed and valuable feedback; our responses are outlined below.
>
> # Limited applicability: the proposed method cannot cover complex tasks that coordinate-based methods offer
> We appreciate the reviewer's insightful comment regarding the limited applicability of S3K-based neural representations when compared to established coordinate-based methods. We fully acknowledge that the current instantiation of our model does not encompass the full range of complex tasks that coordinate-based methods cover.
> Yet, we would like to reiterate and highlight the central goal of our work: **introducing a novel perspective by integrating state-space models into data-specific neural representations**. Our intention is not to claim immediate superiority and applicability over state-of-the-art methods across all tasks; we cannot readily expect to achieve such results in a single step, considering that the coordinate-based models have been continuously developed by numerous researchers over a considerable period. Rather, **we aim to open up a new direction that has been largely unexplored**. Nevertheless, our work shows promising potential by improving reconstruction quality without increasing decoding complexity, and we believe it can be further enhanced through future exploration, opening up a new avenue in this field.
>
> # Can SSM-based methods be extended to create coordinate neural representations (CNR)?
> A straightforward application of SSMs to CNR is not currently feasible due to two fundamental limitations:
>   - SSMs are inherently length-preserving, whereas CNR requires mapping coordinate queries of shape $(N, D)$ to $N$ scalar outputs
>   - Current S3K formulations are not designed to natively handle coordinate inputs.
>
> A possible starting point could be to treat coordinates as multi-dimensional signals and apply 1D SSMs along each axis and then aggregate the resulting features to produce the target value. This idea is conceptually compatible with the continuous-signal modeling properties of SSMs and their inherent compactness suggests potential benefits for CNR. However, these extensions require dedicated investigation; developing a well-founded SSM-based CNR is an open problem that we leave for future work.
>
> # Decoding speed compared to MLP methods such as InstantNGP
> InstantNGP indeed offers faster decoding speed than most MLP models, but requires significant modifications to match comparable performance under constrained representation capacity (Table 1 from [1]): even after more than 40 hours of training, it does not reach PSNR level of 30 with 0.436 bits per pixel (bpp). Note that the baselines from Table 4 achieve ~0.09 bpp and achieve beyond 32dB in PSNR. Accordingly, subsequent MLP-based models adopted InstantNGP [1, 2] have modified its structure to improve its fidelity, while significantly sacrificing its decoding speed (e.g., Table 7 from [1], Table 17 from [2]).
>
> # Additional baselines: NVRC, *-Boost
> NVRC is not included in Figure A3 for three reasons:
>
>   1. NVRC is trained for 720 epochs, whrereas the baselines in Table A5 are all trained for 300 epochs, making a direct comparison unfair.
>   2. Its reconstruction quality for video is not explicitly reported; estimating performance solely from the bpp-fidelity curve (Figure 3 in the NVRC paper) would be unreliable.
>   3. Its source code is not publicly available, preventing us from reproducing its results under matched conditions.
>
> Hence, we are currently training our model for 720 epochs, and have contacted the NVRC authors to obtain their reported performance. We will include the comparison as soon as the corresponding values become available.
>
> *-Boost also improves existing NeRV models, but it does so by modifying their decoder architecture. In contrast, our approach targets the encoder to enhance representation quality without modifying the decoder. Since *-boost is orthogonal to ours, we are conducting experiments to see the effect of applying *-boost to H-NeRV and Ours-H, which share the same base architecture. Please stay tuned; we will share all the results as soon as they are available!

---

> > ### Author Response · Authors · 2025-11-21
> >
> > # Mixed empirical justifications
> >
> > We thank the reviewer for the observation. We would like to clarify a few points regarding the empirical results and efficiency trade-offs:
> >   1. **A single model addressing all three modalities**: S3K-LPNet offers a unified architecture that supports image, video, and 3D reconstruction within one model. This contrasts with MLP-based approaches such as FINER [3], which primarily focus on images and 3D objects, or HiNeRV [4], which targets video reconstruction only. Despite this broader scope, our model achieves a competitive performance on both images (Table A6) and videos (Figure A3), a combination that existing MLP-based methods do not handle simultaneously.
> >   2. **Decoding speed matters for video**: In video reconstruction, decoding speed is critical due to real-time streaming constraints. Our method consistently achieves high reconstruction quality without compromising decoding efficiency, which is often more important than training cost in practical scenarios. Compared to HiNeRV, our reconstruction quality is only 0.5dB lower, but offers 6x faster decoding, offering a meaningful trade-off for real-world applications (Figure A3).
> >   3. **Encoding complexity vs. reconstruction quality**: Higher encoding complexity does not necessarily lead to better reconstruction, as it does not increase the representation capacity (i.e., the size of the input embedding or the decoder parameters). For instance, in our early experiments, stacking additional HNeRV encoder blocks or adding extra layers consistently degraded performance. This suggests that performance gains come from careful architectural design rather than sheer complexity. Our LPNet-S3K demonstrates that a well-structured encoder can improve reconstruction while keeping decoding efficient.
> >
> > # Final remark
> > As a final remark, we would like to gently emphasize that the main focus of our work is to explore the novel integration of state-space models into compressive neural representation frameworks. While the current performance may not yet match that of highly optimized existing approaches—largely due to the limited exploration in acceleration and architectural tuning—we hope the reviewer will recognize our effort in presenting a new, orthogonal direction that has not been previously explored in this domain. We believe this contribution opens up promising avenues for future research.
> >
> > ### References
> >
> > [1] Scalable Neural Video Representations with Learnable Positional Features (Kim et al., NeurIPS 2022)
> >
> > [2] Efficient Neural Video Representation with Temporally Coherent Modulation (Shin et al., ECCV 2024)
> >
> > [3] FINER: Flexible Spectral-Bias Tuning in Implicit Neural Representation by Bariable Periodic Activation Functions (Liu et al., CVPR 2024)
> >
> > [4] HiNeRV: Video Compression with Hierarchical Encoding-based Neural Representation (Kwan et al., NeurIPS 2023)

---

> > ### Author Response · Authors · 2025-12-03
> >
> > ### **vs. NVRC**
> >
> >
> >
> > After 720 epochs, Ours-P achieves 34.76 dB PSNR at 0.023 bpp. Because the NVRC authors have not yet provided their numerical results, we rely on their Fig. 3, which appears to place NVRC around 36 dB at a similar bitrate. Though, it is worth mentioning that its decoding speed is reported as 21FPS, whereas ours achieves 15x faster speed on inferior GPU hardware (NVRC: NVIDIA RTX 4090, Ours: RTX 6000 ADA).
> >
> >
> >
> > | Model  | BPP ↓ | PSNR ↑ | Decoding speed (FPS) ↑|
> > |--------------|--------|-------|-----|
> > | NVRC | $\approx$ 0.02  |$\approx$ **36** | 21 |
> > | Ours-P  | 0.023  | 34.76 | **301** |
> > **Table R2**. Comparison between ours and NVRC.
> >
> >
> >
> > ### **vs. \*-Boost**
> >
> >
> >
> > We conducted an additional experiments comparing the impact of our method, *-Boost, and their combination, summarized in Table R3 below.
> >
> >
> >
> > | Model  | Params ↓ | PSNR  ↑|
> > |--------------|--------|-------|
> > | HNeRV | 3.0M | 37.43 |
> > | Boost-HNeRV  | 3.1M | 41.03 |
> > | Ours-HNeRV | 3.0M | 39.21 |
> > | Boost-Ours-HNeRV | 3.1M | 41.08 |
> > **Table R3**. Effect of ours and *-Boost when applied to HNeRV.
> >
> >
> >
> > Both methods provide clear and consistent improvements over the base model. Although the incremental gain observed when our method is applied on top of Boost-HNeRV is naturally modest, this is expected in the high-PSNR regime: once performance enters the low-40s, most methods encounter a saturation region where further improvements become inherently difficult. Within this challenging range, our method continues to offer stable and complementary benefits.

---

### Official Review · Reviewer_kdKF · 2025-10-31

**Soundness:** 2
**Presentation:** 3
**Contribution:** 2
**Rating:** 4
**Confidence:** 3

**Summary:**

The authors explored the application of state space model (SSM) in visual representation. They analyzed the feasiblity to use an SSM to extract the frequency domain information and designed a pyramidal network structure (S3K). They combined this design with different convolution encoders and verified the performance on multiple input modalities, incouding DVS, 2d image and 3d points. The result shows that their method have some advantages in visual representation quality and extendability to different modalities.

**Strengths:**

- Explored the novel application of SSM in visual representation encoding
- Clearly organized architecture and consistent logic
- Flexibile input forms, covering multi modalities

**Weaknesses:**

- Insufficient analysis on the motivation of method design

  The authors didn't explain the meaning of taking outer product of the 1d convolution kernel into multi dimansional

- Some methods not clearly described

  The article didn't define how to convert an image into the input of the proposed S3K Conv module

- No comparison with populat visual encoding methods like VAE and diffusion

  VAE method is commonly used in visual encoding in various visual and multimodal models. New methods should compare with important baselines and clarify the improvements and advantages

- Format issue in figure caption and referneces

  Figure numbering skipped 2, and the later figure references are mis-marked

**Questions:**

1. In Section 3.1, the authors claimed that SSM can encode a signal into frequency domain (coefficients of a sine series). For this purpose, DFT is the common approach, with mature and efficient algorithmic and hardward implementations. For digital image, there are also DCT and many other methods to yield the frequency domain representation of an image. What are the advantages to use SSM compared with these transformations?
2. In Section 4.2, the authors extend the dimansion of the kernel by making outer product. What is the meaning of this operation in the aspect of State Space Models?
3. What is the relationship between an input (2d image for instance) and the input function $\phi(t)$?
4. What are the advantages of the proposed S3K method compared with the broadly used VAEs?

---

> ### Author Response · Authors · 2025-11-21
>
> We sincerely thank the reviewer for taking their valuable time to provide precious comments. Please find our responses to each point below.
>
> # Figure format issue
> Thanks for pointing out the formatting error. This is currently fixed in the revision.
>
> # Insufficient explanation for the kernel outer product.
> We take the outer product of multiple 1D S3Ks to form an $n$D S3K because this construction directly mirrors how $n$D basis functions are built (Line 294). As implied in Theorem 4.1, applying an 1D S3K is equivalent to projecting the input onto a 1D SSM-derived basis. When extending from 1D to $n$D, the corresponding basis functions become tensor products of 1D bases: e.g., for a 1D basis $\xi(x)$ defined on $[0,W]$ and another 1D basis $\psi(x)$ defined on $[0,H]$, the 2D basis is constructed as $\xi(x)\psi(y)$ defined on $[0,W] \times [0,H]$.
> Once the basis expands in this manner, the convolution kernels that realize those projections must match that structure. The outer product of the 1D S3K kernels produces exactly the tensor-product basis projection in $n$D: each dimension contributes its own S3K, and the resulting kernel evaluates the joint basis function over all coordinates. This yields the mathematically correct operator and is a computationally natural way to lift 1D S3K projections into higher dimensions.
> We appreciate the reviewer’s perspective and will revise the manuscript with these clearer explanations to make our design choice fully transparent and understandable.
>
> # Advantages of using SSMs over DFT/DCT
> We thank the reviewer for this insightful question. While DFT and DCT provide efficient fixed-basis frequency representations, SSMs distinguish themselves by learning basis functions from data. This allows the network to adapt to the structure of each input, capturing patterns that DFT or DCT cannot, including non-periodic or decaying components. Furthermore, SSMs integrate more naturally with neural networks for end-to-end reconstruction, whereas DFT and DCT often produce high variance in intermediate feature maps [1], which may lead to instability during training. We will clarify this distinction in the revision to make the motivation for using SSMs more explicit, right after addressing all reviewers’ concerns for ease of editing the draft.
>
> # Relationship between an input and the input function $\phi(t)$
> $\phi(t)$ represents the underlying input signal that produces the discrete observations $ \textbraceleft \phi_{n} \textbraceright_{n \in \mathbb{N}} $ (e.g., pixel values) we encounter, following the standard SSM convention where the discrete inputs $\textbraceleft\phi_n \textbraceright_{n\in\mathbb{N}}$ are regarded as datapoints uniformly sampled from a continuous signal $\phi(t)$; i.e., $\phi_n = \phi(n\Delta)$ for a positive step size $\Delta$. For 2D inputs, $\phi(t)$ generalizes to a surface function $\phi(x,y)$, representing the pixel values of an image $\textbraceleft\phi_{w,h} | w = 0,1,\ldots, W-1, h = 0, 1, \ldots, H-1 \textbraceright$, with $\phi_{w,h} = \phi(w\Delta, h\Delta)$.
>
> # VAEs and diffusion models omitted for comparison
> In the context of data-specific neural representation, we assume the reviewer’s reference to “VAE” as either (1) probabilistic models used in neural image compression (e.g., [2]) or (2) large-scale VAE-style autoencoders such as Flux [3].
> We are still unsure which specific diffusion model the reviewer refers to, and we would appreciate clarification so that we can evaluate it more properly.
>
> For both categories, we believe VAEs are not appropriate baselines for our setting, for the following reasons.
>
> **Model size is fundamentally incomparable**: VAEs achieves compression by modeling the data distribution, either through entropy-based coding or via a learned low-dimensional latent space. This requires very large networks, often with hundreds of millions of parameters, far beyond the model sizes considered in our setting. Our work, by contrast, focuses on data-specific representations that reconstruct a single sample using a lightweight network, without entropy models, learned priors, or external codebooks. For fairness, we therefore restrict baselines to methods that reconstruct inputs solely from the parameters learned from the input itself.
>
> **Compression ratio lies on a completely different scale**: In data-specific neural representations, every parameter is dedicated to encoding a single target sample, yielding significantly stronger compression ratio than VAEs: e.g., the bits-per-pixel to achieve 30> PSNR required for Flux is 2.33 whereas that of models in Table 4 is 0.09.
>
> Given these fundamental differences in training scheme and scale, VAEs are not an appropriate baseline for this context.
>
>
> ### References
>
> [1] Spectral Representations for Convolution Neural Networks (Rippel et al., NeurIPS 2015)
>
> [2] End-to-end Optimized Image Compression (Balle et al., ICLR 2017)
>
> [3] Flux (Black Forest Labs, 2023)

---

### Official Review · Reviewer_ADR1 · 2025-10-31

**Soundness:** 3
**Presentation:** 4
**Contribution:** 3
**Rating:** 6
**Confidence:** 2

**Summary:**

This paper explores the integration of state-space models (SSMs) into data-specific neural representations (e.g., implicit neural representations for images, videos, and 3D objects). The authors first analyze the theoretical foundation of SSMs, showing that their latent states effectively capture the coefficients of continuous basis functions and thus suit compact signal modeling. Based on this insight, they propose a new Structured State-Space Kernel (S3K), which translates SSM dynamics into convolutional kernels, enabling multi-dimensional processing and inherent downsampling.

The final architecture, LPNet-S3K, integrates S3K layers into a Laplacian pyramid structure and demonstrates strong performance across diverse modalities—images (Kodak, CLIC2020), videos (Bunny, UVG, DAVIS), and 3D objects (Objaverse). The results show consistent gains over ConvNeXt- and Mamba-based baselines and over established NeRV-family models, validating both theoretical and empirical advantages.

**Strengths:**

1. Instead of simply integrating the off-the-shelf state-space model, this paper provides the first systematic theoretical and empirical exploration of applying SSMs to data-specific neural representations. The analysis linking SSM parameters to signal reconstruction coefficients offers a new, signal-processing–grounded perspective on neural compression. In particular, it found a non-trivial conclusion that SSMs consistently outperform the transformer in data representation, which is also explained from the perspective of a classical sinusoid signal processing task.

2. The work revisits how SSMs should be incorporated and identifies some design findings: stacking SSMs can harm reconstruction; applying them to downsampled multi-scale features improves quality; and Laplacian pyramid decomposition provides the best balance between redundancy and expressivity.

3. The experiments are broad and convincing—covering 2D, video, and 3D data. LPNet-S3K consistently improves reconstruction quality with comparable model size and decoding speed. Note that performance gains are achieved via encoder modification, keeping decoders identical to baselines.

**Weaknesses:**

While empirical findings (e.g., stacking degradation, Laplacian advantage) are useful, their underlying reasons are only intuitively discussed. A deeper analytical or visualization-based explanation (e.g., spectral bias, basis overlap) would strengthen the claims.

While this paper shows the advantages of state space model in data representations, I encourage the author to build a SOTA implicit neural representation-based neural codec based on it.

**Questions:**

See in weaknesses.

---

> ### Author Response · Authors · 2025-11-21
>
> We thank the reviewer for their thorough review and helpful suggestions, and below we address each point.
>
> # Visual materials to support empirical findings in Sec. 3
> We visualize the intermediate feature maps of the variants we introduced in Sec 3.2.
> You can find the added figures in the current revised version of the manuscript: Figure 3 and Figure 4.
>
> In Figure 3, each feature map is displayed after taking L2 norm across the channel dimension.
> For all experiments, we use Mamba as the SSM block.
> The “Stacked” variant produces a more block-like feature map from the earlier layer and exhibits clear artifacts in the final output. Regarding that the final feature map of every variant resembles each other, the artifact of the “Stacked” variant’s output image can be understood as arising from the amplified artifacts accumulated from its intermediate layers, supporting our hypothesis of generation loss (Line 203).
>
> We also visualize the outputs of SSM blocks for both the “Image pyramid” and “Laplacian pyramid” variants in Figure 4. Note that these two variants share the same architecture; the only difference lies in their input decomposition. We extract the output SSM feature maps and compute a correlation matrix between them.
> The correlation analysis reveals that features extracted from the Laplacian pyramid are less redundant, showing lower correlation across channels. In contrast, the Image Pyramid features exhibit higher similarity, indicating more overlap and less efficient decomposition. This supports the view that Laplacian decomposition encourages more independent, informative feature extraction.
>
> # Building a SOTA implicit neural representation-based codec
>
> We appreciate the reviewer’s suggestion to develop a full SOTA INR-based neural codec. This is indeed the next step of our research agenda (Line 503). We would like to clarify, however, that constructing a decoder that fully leverages the representation structure induced by an SSM-based encoder is itself a substantial contribution. Specifically, the decoder must leverage the basis implicitly learned within each S3K layer of the encoder, and these bases are distributed across multiple S3K-convolutional layers, making it challenging to correctly and mathematically assign them during decoding. Although Theorem 4.1 provides a theoretical pathway to invert the S3K representation, this formulation requires multiple matrix inverses and divisions, which are numerically unstable and practically infeasible. Extending this derivation to the 2D case further increases the mathematical and computational complexity.
>
> In short, building a theoretically sound and practically effective decoder for SSM-based INRs is a nontrivial challenge—significant enough to deserve dedicated work—which is why it remains our next major objective.

---

### Official Review · Reviewer_DAda · 2025-10-31

**Soundness:** 3
**Presentation:** 4
**Contribution:** 3
**Rating:** 6
**Confidence:** 3

**Summary:**

This well-written paper proposes using modern state-space models (SSMs) for data-specific implicit neural representations (intentionally overfitting SSMs to a single datum) for image, object, and video data. They introduce a structured state-space kernel (S3K) that turns an SSM into a convolutional kernel, allowing for natural n-D preprcessing and learnable downsampling. The final encoder uses a Laplacian-pyramid design with S3K layers and plugs into existing decoders, and they compare against a rich set of baselines.

**Strengths:**

Clear link between SSMs and continuous INRs through basis expansions.

S3K has a concise definition and extends clearly to nD using 1D outer products, allowing it to potentially drop into many encoder models.

Good ablations studies and architectural insights: they evaluated (a) stacked, (b) image pyramid, and (c) Laplacian pyramid variants (showing the last to be preferable across models). This shows that stacking in SSMs may harm reconstruction (they speculate through repeated projection amplifying artifacts).

They provide a number of ablations, including on input adaptivity, real vs. complex parameters, inverted bottlenecks, and SiLU/RMSNorm.

On video benchmarks, replacing only the encoder improves PSNR.

Theory shows relationship between diagonalizable A and spectrum and supports compression framing.

**Weaknesses:**

Fidelity metrics (PSNR/SSIM) alone don’t substantiate “compact representation.'

The memory/compute overhead of explicit kernels is substantial (the authors acknowledge this).

**Questions:**

Are fidelity metrics (PSNR/SSIM) enough to support compression claims? It would be nice to see, e.g., bitrate comparisons / rate-fidelity curves?

Although acknowledged, the compute/memory overhead is significant. Could the authors implement one of the suggested efficient formulations, at least in one case/example?

Could you compare against SSM-convs that don’t build explicit kernels?

How sensitive are results to state size? Does this work for very high res images/videos on realistic training budgets?

---

> ### Author Response · Authors · 2025-11-21
>
> We truly appreciate the reviewer’s time and detailed insights, and we provide our responses below:
>
> # Fidelity metrics are not enough to support compression claims
> Our motivation for reporting higher PSNR under the same representation capacity (i.e., the number of parameters) was to demonstrate compactness: if two models use the same number of parameters to represent a sample, the one achieving higher PSNR is compressing the sample more efficiently, as it embeds more information into the same-size representation. We agree, however, that this view does not fully capture model behavior under various compression ratios, and we will therefore include full rate-fidelity curves as the reviewer suggested.
> Unfortunately, the baselines in Table 4 do not report their performance across varying model sizes, which is required to obtain multiple bitrate points for the rate-fidelity plot. While Bunny (Table 3) contains several reported values and allows us to plot a curve, that comparison would be limited to a single video and thus insufficient. Therefore, we are currently reproducing their implementations on UVG videos to measure reconstruction quality at different model sizes. This process will take some time, but we will provide the requested rate-distortion curves as soon as the values are obtained.
>
> (edit) Rate–distortion curves for both Bunny and UVG are now included in Fig. 6 of the revised manuscript.
> We control model size by adjusting the intermediate channel width of each model, and apply post-training quantization following HNeRV [5].
> As shown in the figure, our method achieves a consistently better rate–distortion trade-off compared to all baselines.
>
> # Can authors implement one of the suggested efficient formulations, at least in one case or example?
> The efficient implementations we mentioned refer to prior advances in state-space modeling–specifically the progression from LSSL [1] to S4 [2], and Mamba [3]’s adaptation to accommodate selective mechanisms without substantial memory consumption. In both histories, the original formulations introduced expressive but computationally heavy algorithms, and the subsequent efforts exhibited a dedicated algorithm to make them practical at scale. These efficiency directions seem conceptually related to our work and may seem straightforwardly applicable, but correctly modifying their algorithms to operate on ours is challenging enough to constitute a separate paper.
> S4, for example, addresses LSSL’s computational challenge of matrix powers by transforming them into frequency-domain inverse problems via the z-transformation and inverse fast Fourier Transform–steps that demand careful derivation to ensure its correctness and numerical stability. The difficulty and novelty of this contribution were significant enough to warrant a separate paper.
> Mamba follows a similar trajectory: the selective SSM formulation is expressive but expensive, and its efficient version relies on custom CUDA kernels and memory-access patterns tailored to GPU architectures. These are substantial algorithmic and systems-level contributions in their own right.
> Given the scale of modifications required to adapt these techniques to our kernel, developing a full efficient implementation is well beyond what can be reasonably added during the rebuttal period. Nonetheless, we view this as an important and promising direction for future work.
>
> # Could you compare against SSM-convs that don’t build explicit kernels?
> We appreciate the reviewer’s question and interpret it as a request to compare the efficiency of our explicit-kernel formulation against SSM-based convolutions that avoid kernel materialization (e.g., S4D, Mamba; we call them implicit SSMs for future reference), since this directly relates to memory and compute behavior.
> S4D [4] and Mamba [3] indeed do not form explicit kernels, and therefore save memory by applying the state transition implicitly. To address the reviewer’s concern, we benchmarked these approaches under the same setting by replacing the convolution in our architecture with an S4D or Mamba block followed by a standard convolution layer. The results are as follows.
>
> | Method    | FLOPs (G) ↓ | Memory (MB) ↓ | Time (ms) ↓ | PSNR ↑ |
> |---|-------|---------|---|-------|
> | S4D + Conv |**2.00** | **453.3** | 100.4 | 26.73 |
> | Mamba + Conv  | 2.36 | 667.3 | 94.4 | 27.51 |
> | S3KConv    | 3.34 | 3445.8 | **62.4** | **28.09** |
> **Table R1**. Efficiency comparison on Kodak, including S3K and other SSMs.
>
> These results highlight the trade-offs: implicit SSM-convs (S4D, Mamba) reduce memory relative to our explicit kernel, but require additional runtime due to the sequential structure of their implicit state updates. Conversely, S3KConv incurs higher memory due to explicit kernel construction, but benefits from more parallel, convolution-like execution and achieves better reconstruction quality.
>
> ### References
>
> [5] HNeRV: A Hybrid Neural Representation for Videos (Chen et al., CVPR 2023)

---

> > ### Author Response · Authors · 2025-11-21
> >
> > # How sensitive are results to state size?
> > We use a state size of 8 (images), 32 (3D data), 64 (videos},  and apply the same state size across all S3K layers for each setting. Empirically, increasing the state size in the first S3K layer meaningfully enhances the performance, likely because this layer is the primary opportunity to capture high-frequency spatial details and a larger state size helps preserve this. In contrast, increasing the state size in later layers yields only marginal gains. For simplicity and consistency, we adopt the first layer’s optimal state size for all layers. In our experiments, performance improves as the state size increases but begins to saturate once the state size is sufficiently large relative to the signal complexity. In practice, this occurs around 8 for images, around 32 for 3D data, and around 64 for videos–values we therefore use throughout the model.
> > We will add the complete ablations on state size as soon as we finish the experiment.
> >
> > # Does this work for very high resolution images/videos on realistic training budgets?
> > Yes—within practical limits. Our experiments already include the CLIC2020 dataset, which contains 2K‐resolution images (2048×1080), and our model trains on these without difficulty. This demonstrates that S3K-LPNet is capable of handling high-resolution inputs under standard training budgets. From 4K-resolution images (3840 x 2160), LPNet-S3K exceeds our 48 GB memory budget.
> >
> > ### References
> > [1] Combining Recurrent, Convolutional, and Continuous-time Models with Linear State-Space Layers (Gu et al., NeurIPS 2021)
> >
> > [2] Efficiently Modeling Sequences with Structured State Space (Gu et al., ICLR 2022)
> >
> > [3] Mamba: Linear-Time Sequence Modeling with Selective State Spaces (Gu et al., COLM 2024)
> >
> > [4] On the parameterization and initialization of diagonal state space models (Gu et al., NeurIPS 2022)

---

> > ### Comment · Reviewer_DAda · 2025-11-26
> > **Response to revisions**
> >
> > I thank the authors for their careful answers to their questions. I am satisfied with their revisions and will increase my score accordingly.

---

> ### Author Response · Authors · 2025-11-28
>
> Thank you very much for the careful assessment of our work and for raising your score—we truly appreciate it. Additionally, we have added the state-size ablation study in the current version of revision to address your earlier concerns more thoroughly (Table A5-A7).
>
> Again, we are grateful for your time and thoughtful comments, which have greatly contributed to refining the manuscript.

---

### Author Response · Authors · 2025-12-03
**Summary of Key Contributions and Rebuttal Process**

Dear Area Chair,

Thank you for taking the time to review our work. Throughout the rebuttal period, we have carefully conducted additional experiments and provided detailed clarifications to address the points raised. We also want to clearly state that we were not involved in any aspect of the recent OpenReview security incident; we fully adhere to the ICLR Code of Conduct and have not engaged in any behavior that would compromise the integrity of the review process. For convenience, we summarize our work and the rebuttal process as below.

### **Our contribution**

Our  primary contribution is the first integration of state-space models (SSMs) into data-specific neural representation, which has never been explored in the field. We provide both theoretical support and empirical exploration to verify its efficacy.

S3K-LPNet, our implementation of this idea, seamlessly integrates with various convolution-based neural representations and is applicable to diverse data modalities (i.e., image, video, 3D), improving reconstruction quality without adding inference cost.

### **Major reviewer concerns and how they are addressed**

We group the reviewers’ major concerns into 4 categories, and provide our response for each as follows. (For brevity, individual concerns and our responses are detailed beneath this message separately, where each concern is labeled from Q1 to Q17 for convenience.)

-   **Lack of evidence to support compression claims and empirical observations (DAda, ADR1; Q1, Q6)**

	We added full rate-distortion curves (Fig. 6), which confirm that our model consistently surpasses baselines in terms of compression. Moreover, we have included visualization-based explanations that support our claims in Sec. 3 (Fig. 3 and Fig. 4 in particular) as Reviewer ADR1 requested.


-   **Comparison against additional or alternative baselines (DAda, kdKF, My7V; Q3, Q12, Q16)**

	We compared our method against NVRC and *-Boost (Q16), as well as a reduced version of our work (Q3, Table R1) as asked by the reviewers. Across these comparisons, our method is either shown to be more effective or to offer a favorable trade-off between reconstruction quality and memory/speed (e.g., only a 1.3dB drop in reconstruction quality while achieving 15x faster decoding compared to the current best method). We also clarified that VAEs and diffusion models are inappropriate baselines in the context of data-specific neural representation, given their substantially larger model sizes and fundamentally different compression regimes (Q12).


- **Potential enhancements (DAda, ADR1, My7V; Q2, Q7, Q13)**

	We acknowledge that our method has room for improvement (acknowledged in Sec. 6, Sec. A.8.4): it can be made more efficient (Q2), it does not yet include a decoder tailored to fully leverage its structure (Q7), and does not currently cover the full range of tasks that coordinate-based approaches can address (Q13). Our primary goal, however, is to establish a new direction by bringing state-space models into neural representations, and we believe this contribution opens promising avenues for future research.

- **Clarification of technical aspects of S3K and their formulation (kdKF; Q9, Q10)**

    We clarified that the kernel outer product stems from a multi-dimensional basis function’s formulation (Q9, Line 294). The interpretation of the input is specified in Eq. (7) and Line 266 (Q10).

**Note**: Reviewer DAda updated their overall score from 6 to **8 (confidence 4)** after the rebuttal, indicating that their concerns were fully addressed.

We direct the AC to the full list of 17 concern-specific details included at the end of this message ("Concise answers for all 17 concerns"), should further context be helpful.

### **Concerns on the review quality (kdKF)**
We would like to respectfully note some concerns regarding the quality of Reviewer kdkF’s evaluation.
-   Several of the reviewer’s questions—such as asking for the meaning of the kernel outer product (Q9) or how the input function is implemented as a convolution kernel's input (Q11)—are about concepts that are already clearly defined in the manuscript (Line 294, Eq. (7)), and reflect fundamental technical misunderstandings rather than issues with our presentation.

-   The reviewer also suggested VAEs and diffusion models as baselines without any references, despite their vastly different model sizes and compression ratios, which makes them inappropriate comparisons in this context.

Notably, these concerns were not raised by any other reviewers, all of whom rated the clarity of our presentation as **4: excellent**, which leads us to question whether Reviewer kdkF engaged with the work at an appropriate level of expertise.

---

Thank you again for your effort and for your dedicated service to the ICLR community.

Authors

---

> ### Author Response · Authors · 2025-12-03
> **Concise answers for all 17 concerns (Q1-Q17)**
>
> ### **(Reviewer DAda)**
>
>
>
> Q1. **Fidelity metrics are not enough to support compression claims.**
>
>
>
> We added the requested rate-distortion curves in the revision; our model clearly surpasses other baselines.
>
>
>
> Q2. **Can authors implement one of the suggested efficient formulations, at least in one case or example?**
>
>
>
> These extensions go beyond what can reasonably be completed during rebuttal. We view them as promising directions for follow-up work.
>
>
>
> Q3. **Could you compare against SSM-convs that don’t build explicit kernels?**
>
>
>
> Shown in Table R1. SSM-convs that do not build explicit kernels are memory-efficient, while ours shows faster inference and higher reconstruction quality at the cost of memory.
>
>
>
> Q4. **How sensitive are results to state size?**
>
>
>
> We added the related ablations in Tables A5-A7. The state size of the first S3K layer is the most influential; later layers show diminishing sensitivity.
>
>
>
> Q5. **Does this work for very high resolution images/videos on realistic training budgets?**
>
>
>
> We train on CLIC2020 (which contains 2K images) without issues, indicating good scalability under realistic budgets.
>
>
>
> ---
>
>
> ### **(Reviewer ADR1)**
>
>
>
> Q6. **Visual materials to support empirical findings in Sec. 3**
>
>
>
> We added visualizations in the revision (Fig.3, Fig.4) that support empirical findings in Sec.3.
>
>
>
>
>
> Q7. **Building a SOTA implicit neural representation-based codec**
>
>
>
> This is a substantial challenge, acknowledged in our limitations. While beyond the rebuttal timeframe, it remains a key direction of our ongoing work.
>
> ---
>
> ### **(Reviewer kdkF)**
>
>
>
> Q8. **Figure format issue**
>
>
>
> Resolved in the revised manuscript.
>
>
>
> Q9. **Insufficient explanation for the kernel outer product**
>
>
>
> It follows the standard definition of constructing a 2D basis via the outer product of two 1D basis functions. Mentioned in Line 294.
>
>
>
> Q10. **Advantages of using SSMs over DFT/DCT**
>
>
>
> SSMs learn data-adaptive basis functions; DFT/DCT are fixed transformations and can introduce high-variance feature maps. Also, learnable bases integrate more naturally into neural network pipelines.
>
>
>
> Q11. **Relationship between an input and the input function $\phi(t)$**
>
>
>
> $\phi(t)$ represents the underlying signal generating the discrete observations. For images, it generalizes to $\phi(x,y)$, where sampled values correspond to pixel intensities.
>
>
>
> Q12. **VAEs and diffusion models omitted for comparison**
>
>
>
> These models are substantially larger and achieve far worse compression ratios; thus they are unsuitable baselines for this setting.
>
> ---
>
> ### **(Reviewer My7V)**
>
>
>
> Q13. **Limited applicability: the proposed method cannot cover complex tasks that coordinate-based methods offer**
>
>
> We acknowledge the current scope but emphasize that our goal is to introduce SSMs as a new pathway for data-specific neural representations. The method should be viewed as a prototype establishing feasibility, with broad room for extension.
>
>
> Q14. **Can SSM-based methods be extended to create coordinate neural representations (CNR)?**
>
>
> While possible in principle, there are fundamental limitations to address. A rigorous SSM-based CNR is an important direction for future work.
>
>
>
> Q15. **Decoding speed compared to MLP methods such as InstantNGP**
>
>
> The baselines in Table A5 already include InstantNGP-style advances, yet still decode slower than convolution-based methods.
>
>
>
> Q16. **Additional baselines: NVRC, \*-Boost**
>
> Compared to NVRC, our Ours-P model reaches 34.76 dB at 0.023 bpp, while NVRC’s curve (Fig. 3 of the NVRC paper) appears around 36 dB at a similar bitrate with 15x slower decoding speed (Table R2). For *-Boost, Table R3 shows that both methods significantly improve HNeRV, and that the combined model still yields gains despite the natural saturation that occurs in the low-40 dB regime.
>
>
>
> Q17. **Mixed empirical justifications**
>
> Our method provides a unified architecture applicable to three modalities (i.e., image, video, 3D) while improving performance without compromising decoding speed, which is particularly crucial for videos. The claim that stronger encoding leads to better performance is nontrivial, as the representation capacity does not increase.

---

### Meta-Review · Area_Chair_8Uep · 2026-01-05

**Summary:**

This work proposes using SSMs to generate data-specific representations. The reviewers agreed that the paper is clear, proposes a novel neural network architecture for data-dependent neural representations with both empirical and theoretical support. The reviewers expressed several concerns, including:
- need for more conclusive empirical evidence to support the authors claims (compact representations, etc.)
- computational efficiency
- some methodologies are not clearly explained
- the model is not SOTA
- comparison with other methodologies
- limited applicability

**Reviewer Concerns:**

The authors addressed several of the reviewers concerns, including providing further empirical support of their claims and explaining methodologies that were not clear. The authors did not run further experiments to address the computational efficiency of their model, compare with other models, expand the applicability or their model, nor optimize to make their model SOTA. The authors argued that these extensions are beyond the scope of the current paper.

**Reviewer Scores:**

I believe that Reviewer DAda would have increased their score based on the authors' response.

Some but not all of Reviewer ADR1's concerns were addressed. I do not think they would have increased their score.

I believe that most of Reviewer kdKF's concerns were addressed and so they would increase their score.

I think that Reviewer My7V's concerns remain and they would not have increased their score.

---

### Decision · Program_Chairs · 2026-01-26

Accept (Poster)